# Using B isotopes and B/Ca in corals from low saturation springs to constrain calcification mechanisms

M. Wall[1,2], J. Fietzke [1], E.D. Crook[3,4] & A. Paytan[4]

Ocean acidification is expected to negatively impact calcifying organisms, yet we lack understanding of their acclimation potential in the natural environment. Here we measured geochemical proxies ($\delta^{11}B$ and B/Ca) in *Porites astreoides* corals that have been growing for their entire life under low aragonite saturation ($\Omega_{sw}$: 0.77–1.85). This allowed us to assess the ability of these corals to manipulate the chemical conditions at the site of calcification ($\Omega_{cf}$), and hence their potential to acclimate to changing $\Omega_{sw}$. We show that lifelong exposure to low $\Omega_{sw}$ did not enable the corals to acclimate and reach similar $\Omega_{cf}$ as corals grown under ambient conditions. The lower $\Omega_{cf}$ at the site of calcification can explain a large proportion of the decreasing *P. astreoides* calcification rates at low $\Omega_{sw}$. The naturally elevated seawater dissolved inorganic carbon concentration at this study site shed light on how different carbonate chemistry parameters affect calcification conditions in corals.

[1] GEOMAR Helmholtz-Centre for Ocean Research Kiel, Marine Geosystems & Marine Ecology, Wischhofstr 1-3, 24148 Kiel, Germany. [2] Department of Palaeontology, University of Vienna, Althanstraße, 1090 Vienna, Austria. [3] Department of Earth System Science, University of California, Irvine, Croul Hall, Irvine, CA 92697-3100, USA. [4] University of California, Earth and Marine Science Building, 1156 High Street, Santa Cruz, CA 95064, USA. Correspondence and requests for materials should be addressed to M.W. (email: mwall@geomar.de)

O cean acidification is projected to lead to negative effects on calcifying organisms, particularly tropical corals[1–3]. Our understanding of the potential fate of corals in the face of changing $pCO_2$ in the ocean is based primarily on controlled laboratory studies (e.g. refs. [4,5]), mesocosm studies mimicking coral community composition[6–8], alkalinisation versus carbon dioxide-enrichment studies in natural coral reef sites[9,10], and a number of field studies with naturally reduced calcium carbonate saturation state ($\Omega_{arag}$)[1,11–14]. These efforts have provided strong evidence that the calcification rates of a large number of coral species investigated to date will decline in response to projected $pCO_2$[15]. However, some studies also report that certain coral species were able to maintain high calcification rates or even benefit from elevated $pCO_2$[1,16–18], suggesting a high resilience potential of some coral species to changing carbonate chemistry[19]. Specifically, the ability of an organism to control the biomineralization process clearly determines its ecological and physiological success under reduced pH conditions[14]. The process of calcification in corals is linked to their ability to control the pH at the site of calcification ($pH_{cf}$) by removing protons out of the calicoblastic space between the tissue and skeleton, where calcification takes place[5]. This enables corals to sustain $pH_{cf}$ well above seawater pH ($pH_{SW}$)[5,19,20]. The physiological capacity of corals to control $pH_{cf}$ may alleviate the decline in coral growth and increase coral resilience to future climate change[19]. Knowledge about internal calcifying fluid $pH_{cf}$ in corals has been derived from a few direct measurements under the calcifying cell layer either using microsensors[21,22] or pH-sensitive dyes[5,23]. These studies confirmed an elevated $pH_{cf}$ of between 0.4 and 2 pH units above ambient seawater in the calicoblastic space. Indirectly, boron isotopes ($\delta^{11}B$) of coral skeletons, which represent the $pH_{cf}$ of the calcifying solution, also suggest an elevated $pH_{cf}$ (e.g. refs. [19,24–26]). Boron isotopes are more readily accessible compared to direct measurements and have the additional benefit that they integrate $pH_{cf}$ history over longer time periods[19,20,24,27]. Studies suggest that $pH_{sw}$ is an important driver affecting $pH_{cf}$[25,28]. However, it was recently demonstrated that changes in seawater dissolved inorganic carbon (DIC) or total alkalinity (TA) can also affect $pH_{cf}$ regulation[23]. Using B/Ca as a proxy for internal carbonate ion concentration ($CO_3^{2-}_{cf}$), provided geochemical evidence that corals can also modulate and adjust the internal DIC ($DIC_{cf}$) concentration. Together—the potential to upregulate $DIC_{cf}$ and $pH_{cf}$—allows for higher carbonate ion concentrations at the site of calcification and hence a higher $\Omega_{cf}$ that facilitates calcification[29,30].

Over the last decade, a growing body of literature has provided evidence that corals subjected to daily and seasonally fluctuating environmental conditions are able to exert a stronger control over their internal physiological attributes, potentially allowing them to better cope with future changes (reviewed in ref. [31]). For instance, in situ flume experiments mimicking natural (daily, seasonal) fluctuating conditions coupled with future $pCO_2$ conditions showed that corals from acidified treatments could maintain a constant calcification $pH_{cf}$ irrespective of changes in seawater $pH_{sw}$[27]. The authors argued that the fluctuating conditions the corals were exposed to likely favour this strong control on internal conditions. That year long experiment, however, cannot tell whether corals can maintain such strong control when exposed to reduced mean seawater aragonite saturation state ($\Omega_{sw}$) for their entire life span. Corals living for their entire life under continuously low $\Omega_{sw}$ and variable environmental conditions can be used to test whether corals can maintain $pH_{cf}$ homoeostasis over long time spans in their natural ecosystem with its complex biological interactions. Because many natural ocean acidification sites also show strongly fluctuating conditions[11,32,33] these settings may be ideal for testing the relationship between environmental variability and acclimation potential of corals to low $\Omega_{sw}$[34].

Here we measured geochemical proxies in the upper most recently formed skeletal parts of *Porites astreoides* corals that were collected along a natural aragonite saturation gradient at submarine springs (locally known as ojos) in Puerto Morelos, Mexico[11]. These geochemical proxies ($\delta^{11}B$-derived estimates of $pH_{cf}$ and B/Ca derived estimates of $CO_3^{2-}_{cf}$) allowed us to infer carbonate chemistry conditions at the site of calcification, which provide valuable new insights into the internal calcification regulation mechanisms in corals exposed to persistent low $\Omega_{sw}$, as well as fluctuating carbonate chemistry conditions[33]. Our results, combined with bio-inorganic calcification models[19,30], identified critical regulation mechanisms and the inability of corals to fully acclimate to these conditions and sufficiently elevate their $\Omega_{cf}$ to sustain growth rates similar to the same species of corals growing at ambient $\Omega_{sw}$.

## Results

**Natural conditions at the ojos.** We used 12 cores from the coral *P. astreoides*: 5 cores collected from the centre of the low $\Omega_{sw}$ ojos and 7 from control present-day $\Omega_{sw}$ sites adjacent (within a few meters) to the ojos[11]. *Porites astreoides*, the species used in this study, represents one of only three calcifying coral species found growing within the discharge impacted area, while nine coral species are found nearby under ambient present-day $\Omega_{sw}$. Previous studies indicate that although the abundance of *P. astreoides* was not significantly reduced at the low $\Omega_{sw}$ ojos, its growth rate (measured as net calcification) decreased significantly by 37% compared to the same species collected at control sites[11]. The control sites have a relatively consistent $\Omega_{sw}$ (on average: $3.92 \pm 0.03$ sd) year round compared to the ojos where $\Omega_{sw}$ is always <2 and ranges from 0.77 to 1.85 (on average: $1.49 \pm 0.14$ sd, Supplementary Table 1 [11]).

**Skeletal $\delta^{11}B$ and thus $pH_{cf}$ as a function of $\Omega_{sw}$.** The $\delta^{11}B$ in the 12 corals analysed ranged from 23.1‰ to 27.6‰, with slight but significantly lower values for corals affected by the ojo discharge where $\Omega_{sw}$ was low (Fig. 1a, Supplementary Table 2; *t*-test: $p = 0.022$; $r^2 = 0.29$, $p = 0.04$). These $\delta^{11}B$ values translate into $pH_{cf}$ that are slightly lower (but not statistically significant) in the corals from sites with $\Omega_{sw} < 2$ with an average internal $pH_{cf}$ of 8.46 ($\pm 0.03$ sem) compared to 8.54 ($\pm 0.01$ sem) at the control sites (*t*-test: $p = 0.085$, Fig. 1b, Supplementary Table 2). The $pH_{cf}$ difference between the corals is relatively small (0.08 pH units) compared to the difference in environmental seawater $pH_{sw}$ of ~0.54 pH units between the sites. Hence, compared to $pH_{sw}$ in their surrounding environment, corals at the ojo centres maintained a higher pH gradient between seawater and the calcifying fluid ($\Delta pH$) in comparison to corals at control sites (Fig. 1c, Supplementary Table 2; *t*-test: $p = 0.002$, $r^2 = 0.89$, $p < 0.001$).

**Skeletal B/Ca, thus $DIC_{cf}$ and $CO_3^{2-}_{cf}$ as a function of $\Omega_{sw}$.** Changes in coral skeletal B/Ca were determined along with $\delta^{11}B$. This ratio varied between 442 and 721 μmol mol$^{-1}$ and did not significantly correspond to $\Omega_{sw}$ (Fig. 2a, $p = 0.86$). Using the $\delta^{11}B$ and B/Ca skeletal proxies together to constrain the carbonate system at the site of calcification suggests an elevation of $CO_3^{2-}_{cf}$ not only due to shifts in internal $pH_{cf}$ but also due to an increase in $DIC_{cf}$ (Supplementary Table 2, Fig. 2b). The ratios of $DIC_{cf}/DIC_{sw}$ —a measure of the upregulation of $DIC_{cf}$ compared to seawater—were significantly higher at the control sites than at the low $\Omega_{sw}$ sites (Fig. 2c, Supplementary Table 2; $p = 0.036$, $r^2 = 0.33$, $p = 0.029$). This is mainly due to the higher than ambient $DIC_{sw}$ (Supplementary Table 1) at the ojos because the $DIC_{cf}$ did

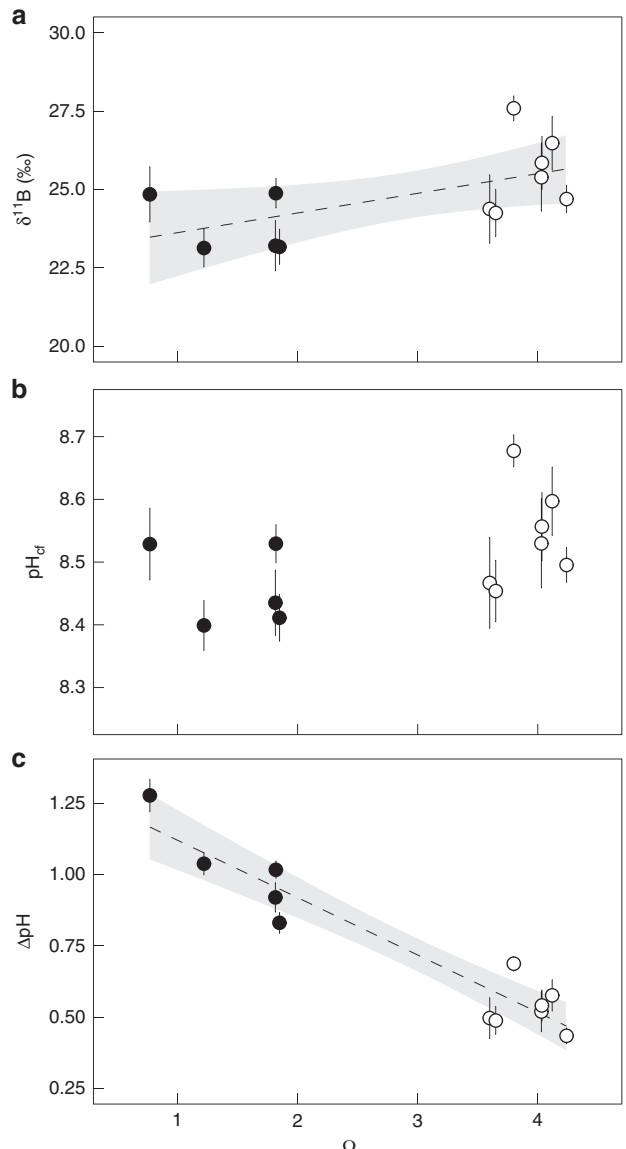

**Fig. 1** *Porites astreoides* internal $pH_{cf}$ regulation based on skeletal proxies. Coral skeletal $\delta^{11}B$ signature **a** from naturally different seawater aragonite saturation state ($\Omega_{sw}$) sites were translated into **b** internal calcifying fluid pH ($pH_{cf}$) and **c** pH up-regulation intensity ($\Delta pH$). Circles represent values for each individual coral colony (mean ± confidence interval). Filled and non-filled symbols denotes the different locations: filled are the centres of the ojos with lower $\Omega_{sw}$ and non-filled the control high $\Omega_{sw}$ site. Dashed line represents regression line for site-specific significant different values in $\delta^{11}B$ and $\Delta pH$ with $\Omega_{sw}$ ($r_{adj}^2 = 0.29$, $p = 0.04$, $r_{adj}^2 = 0.88$, $p < 0.001$, respectively) and grey area denotes the 95% confidence band. Individual values are mean ± 95%-confidence interval

not vary significantly between the sites (Supplementary Fig. 1, Supplementary Table 2; $p = 0.85$).

**Changes in calcification conditions and calcification rates**. We applied a bio-inorganic model (IpHRAC model from McCulloch and colleagues[19], see the "Methods" section) to calculate average $\Omega_{cf}$ and associated relative calcification rates using our proxy data. We used the combined $pH_{cf}$ and the B/Ca-derived $DIC_{cf}$ concentration at the site of calcification for corals from both ojo and control sites. With this bio-inorganic model both $\Omega_{cf}$ (from $16.1 \pm 0.3$ sem to $13.1 \pm 0.7$ sem, Fig. 3a; $p = 0.042$) and

calcification rates (from $1.0 \pm 0.09$ sem to $0.54 \pm 0.09$ sem; $p = 0.036$) decreased with decreasing $\Omega_{sw}$ and correlated well with the observed calcification response previously reported for these corals[11] (Fig. 3b). 41% of the variation in measured net calcification rates can be explained by internal changes in $\Omega_{cf}$ derived from the geochemically determined calcifying conditions (Fig. 4a; $r_{adj}^2 = 0.41$, $p = 0.011$). The internal calcifying fluid parameters are clearly distinct between corals from the different field sites, strongly indicating a combined effect of $DIC_{cf}$ and $pH_{cf}$ modulation on *P. astreoides* calcification performance (Fig. 4b).

## Discussion

Coral calcification is one of the most fundamental processes in reef ecosystems and is essential for reef accretion and ecosystem diversity; however, calcification may be impacted by changes in seawater carbonate chemistry. Although corals are sensitive to changes in ocean carbonate chemistry[15], the underlying physiological mechanisms that determine vulnerability are far from understood. Natural sites with low aragonite saturation that select for genotypes that can calcify under such conditions and permit decade-long developmental acclimation to changes in $\Omega_{sw}$ are invaluable model systems for understanding the resilience of corals and coral calcification processes. Here we reveal that corals grown for their entire lifetime at low aragonite saturation conditions in their natural environment, at ojos in the Caribbean, exert strong control on both $pH_{cf}$ and $DIC_{cf}$, thereby modulating $CO_3^{2-}_{cf}$, $\Omega_{cf}$, and calcification rate. At the calcification site, both parameters that control $\Omega_{cf}$ ($pH_{cf}$ and $DIC_{cf}$) decreased only slightly along the ambient $\Omega_{sw}$ gradient in which the analysed corals live, highlighting the strong control of *Porites asteroids* corals over the biomineralization process. Yet the combined change in $pH_{cf}$ and $DIC_{cf}$ corroborate the observed decline in calcification rate along the environmental gradient (Fig. 4b). Interestingly, at this field site ojos with low $\Omega_{sw}$ had elevated $DIC_{sw}$, but this did not result in higher $DIC_{cf}$ concentrations in the calcifying fluids, indicating a decoupling of internal and external DIC concentrations. This indicates that corals have significant control over the carbonate chemistry of the calcifying fluid, likely mediated by bicarbonate transporters (NBC, SLC4 family of ion transporters) that are localised in the calicoblastic epithelium, as well as other, not yet identified acid–base relevant transporters[35]. Carbonate chemistry at the calcification site clearly differs between coral growing at the control and ojo locations. The difference explains 41% of the observed difference in calcification rate; however, it still leaves 59% of the variation in calcification rate unexplained (Fig. 4b).

In our study, we took advantage of the inherent conditions of this submarine springs system, including the strong environmental fluctuations and the fact that carbonate chemistry is controlled by saline groundwater discharge, allowing us to provide new facets on drivers of coral calcification in natural settings affected by ocean acidification. In the subsequent discussion we will outline the novel insights we derive from the observed internal carbonate chemistry conditions at this natural low $\Omega_{sw}$ site, discuss potential mechanisms that control calcification rates, add to the ongoing discussion on how seawater carbonate chemistry affects regulations of internal conditions at the site of calcification (e.g. ref. [23]), and emphasise the importance of deciphering internal calcium regulation[36–38].

The ability of organisms to modify $pH_{cf}$ reflects the strong effect of intracellular biological processes on coral calcification and is manifested in skeletal isotopic composition. The control of $pH_{cf}$ represents one mechanism to counter external seawater conditions[39]. The boron isotopic-derived $pH_{cf}$ values we report are similar to those reported in other studies for *Porites*[25–27]. The

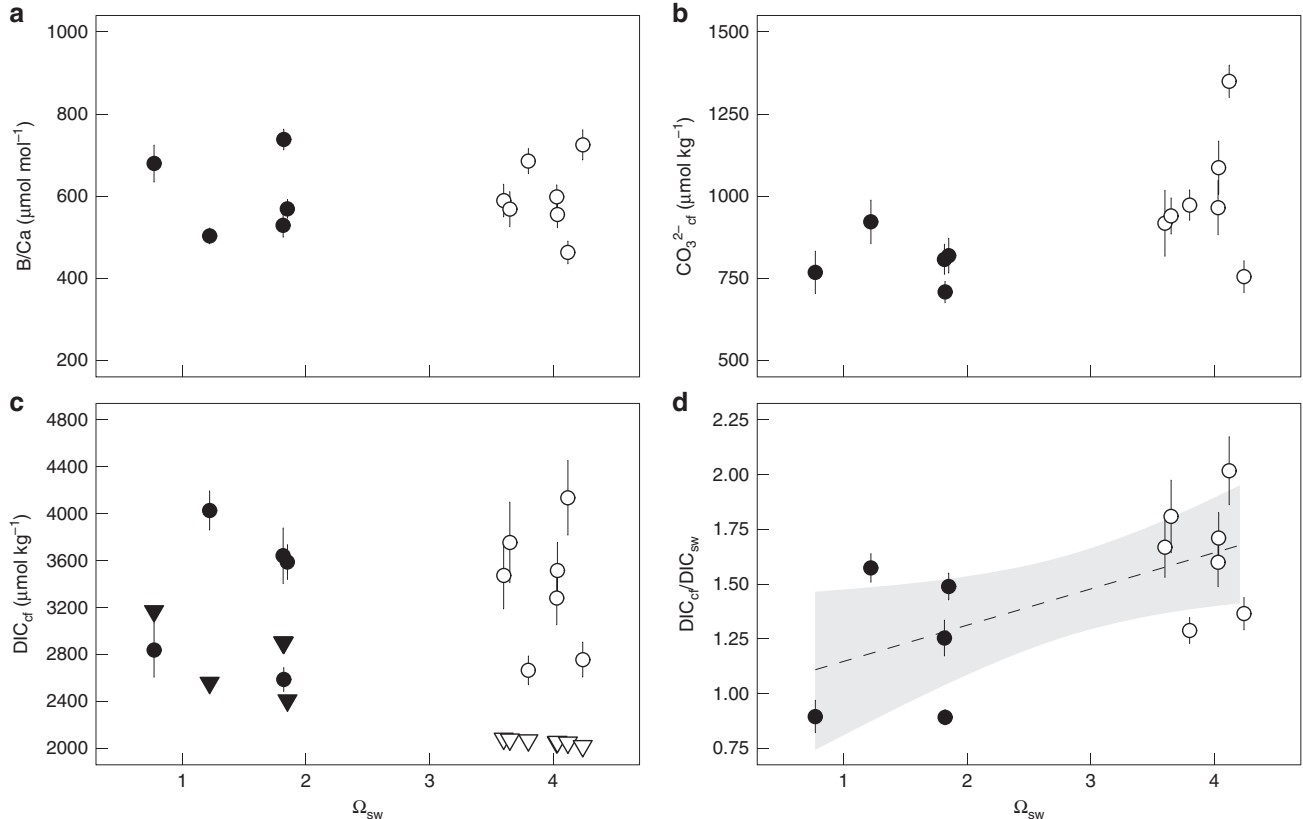

**Fig. 2** *Porites astreoides* internal $CO_3^{2-}$ and $DIC_{cf}$ based on skeletal proxies. Coral skeletal B/Ca ratio **a** from naturally different seawater aragonite saturation state ($\Omega_{sw}$) sites were translated into internal calcifying carbonate ion concentration ($CO_3^{2-}$) values **b** and dissolved inorganic carbon ($DIC_{cf}$) concentration, as well as upregulation compared to seawater ($DIC_{cf}/DIC_{sw}$) (**c**, **d** respectively). Circles represent values for each individual coral colony (mean ± confidence interval). Filled and non-filled symbols denotes the different locations: filled are the centres of the ojos with lower $\Omega_{sw}$ and non-filled the control high $\Omega_{sw}$ site. Triangles in **c** represent seawater $DIC_{sw}$ concentrations. Dashed line represents regression line for site-specific significant different values for $DIC_{cf}/DIC_{sw}$ with $\Omega_{sw}$ ($r_{adj}^2 = 0.33$, $p = 0.029$) and grey area denotes the 95% confidence band. Individual values are mean ± 95%-confidence interval

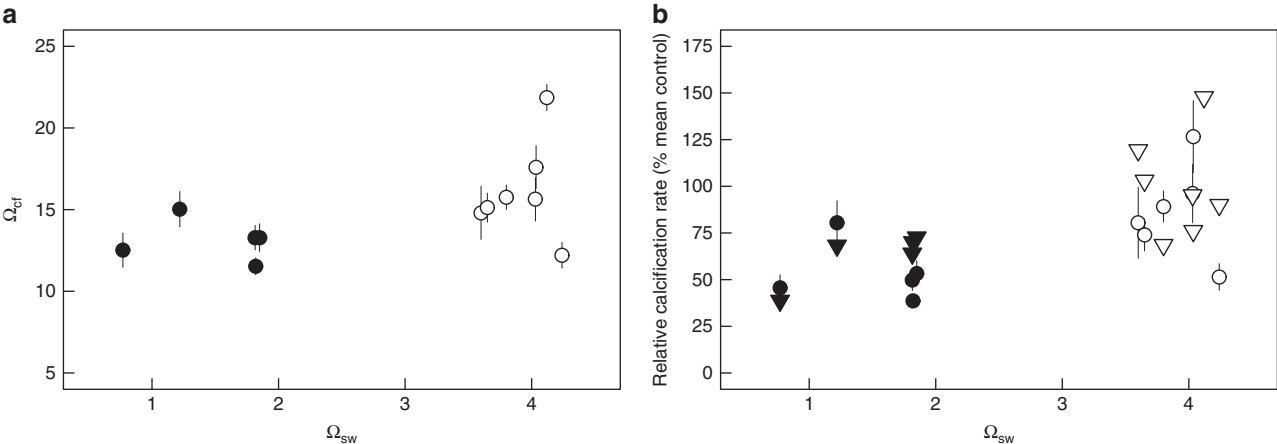

**Fig. 3** Growth response of *Porites astreoides* corals. The modelled growth response displays relative changes in calcification rate (relative calcification rate = mean control/individual colony). Calcification rates were calculated following the IpHRAC model[19] (internal pH regulation and abiotic calcification: Calcification = $k*(\Omega_{cf}-1)^n$) with calcifying fluid aragonite saturation state ($\Omega_{cf}$) was calculated from the average internal calcifying fluid pH ($pH_{cf}$) of individual colonies and the dissolved inorganic carbon ($DIC_{cf}$): in **a** dependent variable $\Omega_{cf}$ is based on $DIC_{cf}$ and $pH_{cf}$ and **b** depicts the respective calculated calcification rates. Circles represent values for each individual coral colony (mean ± confidence interval). Filled and non-filled symbols denotes the different locations: filled are the centres of the ojos with lower seawater aragonite saturation state ($\Omega_{sw}$) and non-filled the control high $\Omega_{sw}$ site. We compared calculated values with measured data[11] (for better comparison also calculated as relative rate, open triangles). Individual values are mean ± 95%-confidence interval

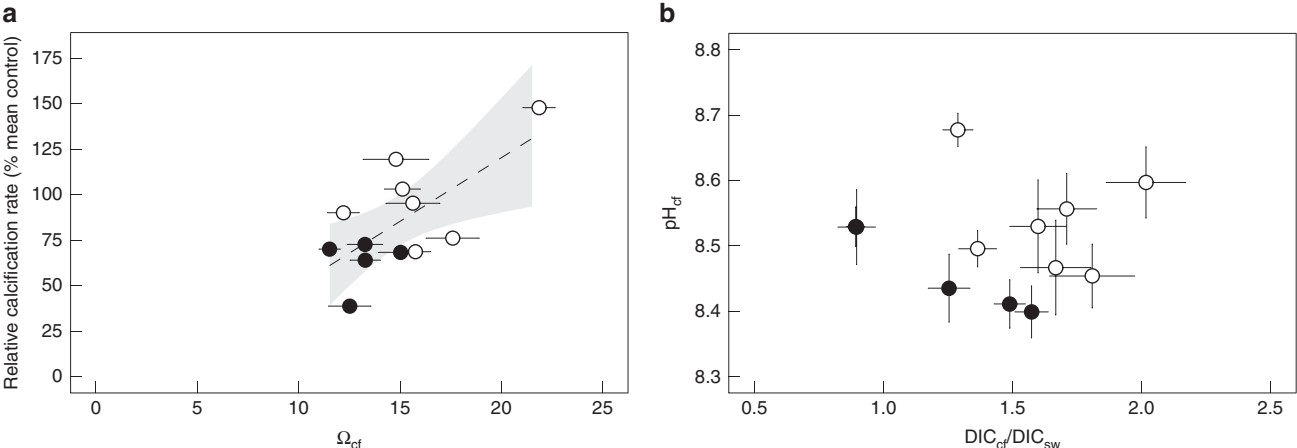

**Fig. 4** Internal calcification conditions in *Porites astreoides*. **a** Calcifying conditions were used to calculate aragonite saturation state at the site of calcification ($\Omega_{cf}$) and related to the measured growth rate of the individual corals[11]. **b** Calcification conditions as internal calcifying fluid pH (pH$_{cf}$) and dissolved inorganic carbon ratio of calcifying fluid to seawater (DIC$_{cf}$/DIC$_{sw}$) was determined for 12 *Porites astroides* coral colonies collected from sites with naturally different seawater aragonite saturation state ($\Omega_{sw}$). Circles represent values for each individual coral colony (mean ± confidence interval). Filled and non-filled symbols denotes the different locations: filled are the centres of the ojos with lower $\Omega_{sw}$ and non-filled the control high $\Omega_{sw}$ site. Dashed line represents regression line for relative calcification rate with $\Omega_{cf}$ ($r_{adj}^2 = 0.41$, $p = 0.015$) and grey area denotes the 95% confidence band. Individual values are mean ± 95%-confidence interval

sensitivity of pH$_{cf}$ to changes in the environmental pH$_{sw}$, however, differed between the different studies, as the environments the corals originated from were distinct[27,40]. All studies observed that pH$_{cf}$ stays within a narrower range (8.2–8.6) compared to large changes in seawater pH$_{sw}$. They all highlight the generally strong control corals exert on pH$_{cf}$. Despite this capacity for regulation, however, the observed pH$_{cf}$ was lower at lower $\Omega_{sw}$ (e.g. refs. [20,25,28,41]). Irrespective of whether corals maintained high pH$_{cf}$, the corals exposed to low $\Omega_{sw}$ maintained a higher proton gradient at lower pH$_{sw}$ (Fig. 2b). A potential driving force that fosters acclimation to various changes a coral may experience is the environmental history corals have been exposed to during their lifetime[31]. For example, pH homoeostasis—the maintenance of internal pH$_{cf}$ irrespective of the external seawater pH$_{sw}$—was observed in corals that live in a highly dynamic naturally variable environment[5,42]. The underlying assumption is that these corals are better able to buffer external changes by exerting a stronger control over the calcifying fluids or by better exploiting times of favourable conditions[27,40]. Although the ojos represent a highly dynamic system[33,43,44], coral performance measured in terms of net calcification was lower at these sites relative to the same species collected at control sites at the same location. Here, life-long exposure to variable and persistently low $\Omega_{arag}$ (<2) did not lead to full acclimation[11]. It is likely that there is a critical $\Omega_{arag}$ threshold beyond which corals are no longer able to fully compensate for external acid–base changes. Such a critical threshold has been observed for corals grown at a Papua New Guinea $CO_2$ seep site, where pH$_{cf}$ homoeostasis was only found for pH$_{sw}$ of >7.8 and $\Omega_{sw}$ of >2.3. Beyond that, pH$_{cf}$ could not reach the same values as those under control conditions, and likely the coral's physiological limit to compensate for changes was reached[40]. This lack of ability to fully compensate for the lower pH$_{sw}$ may be responsible for the slight differences in pH$_{cf}$ observed in this study.

The use of our dual geochemical proxy data to model coral growth (e.g. IpHRAC[19,30]) allowed us to further pinpoint potential mechanisms of how external seawater conditions affect internal calcifying conditions and ultimately skeletal growth. Calcification was once thought to be a passive diffusion process of seawater that brings external DIC to the site of calcification

(potentially gaining DIC from metabolic $CO_2$ by passing through the paracellular pathways)[45] and by active ion transporters[46] that result in an elevation of pH$_{cf}$, thereby facilitating precipitation. At our study sites, DIC$_{sw}$ is significantly higher at the low $\Omega_{sw}$ sites (in average 2790 μmol kg$^{-1}$ compared to control average DIC$_{sw}$ of 2050 μmol kg$^{-1}$) allowing us to decipher the role of external DIC$_{sw}$ in modulating calcification regulation processes. If corals modify internal DIC$_{cf}$ by simply up-regulating DIC$_{cf}$ from the external concentrations baseline, we would expect higher DIC$_{cf}$ values for the ojo corals where DIC$_{sw}$ is higher. Under such assumption the elevated DIC$_{cf}$ compensates for the slightly lower pH$_{cf}$ effect on $\Omega_{cf}$ and calcification rates would essentially be similar between sites (Supplementary Fig. 1). However, our data clearly demonstrate that DIC$_{cf}$ is not directly linked to external concentrations and can differ significantly from that of seawater[22,30,47] (reported DIC$_{cf}$-upregulation values range from 1.6 to 3.2[30,38], with the ojos corals at our sites at the lower end), and this impacts $\Omega_{cf}$ (more precisely $CO_3^{2-}$) and calcification. A recent study under laboratory conditions with *Stylophora pistillata*[23] observed that changes in DIC$_{sw}$ concentration modulates internal pH$_{cf}$ regulation, with higher external DIC$_{cf}$ facilitating higher internal pH$_{cf}$, resulting in a clear correlation between seawater DIC$_{sw}$/H$^+_{sw}$ and pH$_{cf}$. Since DIC$_{sw}$ at the ojos is significantly higher than at the control sites, one could expect this to compensate reduced pH$_{cf}$ up-regulation induced only due to changes in seawater pH$_{sw}$. Yet we do not see a strong correlation between pH$_{cf}$ and seawater DIC$_{sw}$/H$^+_{sw}$, suggesting different drivers for $\Omega_{cf}$ regulation in *P. astreoides* compared to those observed in *S. pistillata*[23]. Nevertheless, the change in pH$_{cf}$ and the limited ability to upregulate DIC$_{cf}$ at the ojos corroborates the observed calcification rate decrease of corals at the ojos. However, these parameters may not be the only drivers for the decline in growth. Recent studies identified internal calcium (Ca$^{2+}_{cf}$) regulation as an additional player in coral calcification responses and emphasised that regulation of Ca$^{2+}_{cf}$ can contribute to a corals' resistance to future ocean changes[36–38,48]. In this sense, the good agreement of our model with the observed calcification response may imply that internal average steady-state calcium concentrations (Ca$^{2+}_{cf}$) at the ojos are lower by some proportion that is related to the pH$_{cf}$ changes, since our model based on pH$_{cf}$ and

$CO_3^{2-}{}_{cf}$ can explain only 41% of the observed calcification decline. This suggests a strong link between $Ca^{2+}{}_{cf}$ and $pH_{cf}$ and supports the idea of a plasma-membrane Ca-ATPase ([49], but see ref. [50]) responsible for $pH_{cf}$ regulation. However, it is possible that $pH_{cf}$ and $Ca^{2+}{}_{cf}$ were both regulated by additional and/or different ion transport mechanisms (e.g. potentially ion exchangers, $Ca^{2+}$-channels)[50,51].

The present study also indicates that the acclimation process in different corals encompass some degree of flexibility in terms of the relative role of $pH_{cf}$ and $DIC_{cf}$ regulation in increasing the $\Omega_{cf}$, with some individuals compensating by adjusting their internal $pH_{cf}$ and others primarily by $DIC_{cf}$ modulation. This may also be true of the role of $Ca^{2+}{}_{cf}$ upregulation. The relative amount, source, and transportation pathways of DIC, $H^+$ and $Ca^{2+}$ to the site of calcification are still not fully understood[52] and transport processes may differ between different coral species and even individual corals of the same species. Another potential driver for the observed differences among studies could be the number and type of symbionts the corals are hosting. Corals at the ojos harbour a higher density of symbionts[53] that may potentially account for the higher energy demands for $pH_{cf}$ up-regulation resulting in the relatively small difference in the internal conditions ($pH_{cf}$, $DIC_{cf}$) we see. Recent work provided the first evidence that coral symbionts (e.g. by modulating the chemical microenvironment within the diffusive boundary layer surrounding the coral that may buffer external changes[54]) and host genotypes can jointly affect coral calcification rates[55]. Similarly, possible interactions with the microbiome (e.g. restructuring of the corals microbiome[56]) or changes in energy acquisition and allocation processes to overcome environmental gradients[57,58] can affect coral growth. Environmental factors may also affect $pH_{cf}$ and $DIC_{cf}$ explaining some of the observed differences between the ojo and ambient corals at our study site. Studies have shown that a decrease in $pH_{cf}$ and $DIC_{cf}$ is associated with increasing temperature[38], yet at our sites the temperatures at the ojos is actually lower, on average, than at control sites. Salinity might also influence regulation processes, yet the measured average values (32.2 psu) as well as the salinity range measured (26–36 psu)[44] at the springs can be tolerated by corals and the long-term exposure to such conditions may have allowed them to develop mechanisms to better cope and adapt to this variable environment[59]. Overall, these environmental and biological parameters may be responsible for the observed internal conditions in the calcifying fluid but likely also affect rates of processes that ultimately affect calcification, and thus contribute to the unexplained component in our relationship between calcification and the geochemically derived $DIC_{cf}$ and $pH_{cf}$. Our geochemical model approach assumes steady-state equilibrium conditions; however, the rates of the various transport processes involved in regulating the chemistry of the calcifying fluid will ultimately dictate the calcification response[60]: these rates may differ between individual coral genotypes, further contributing to the offsets between the model output and observations.

In this study, we utilised a dual geochemical proxy approach ($\delta^{11}B$ and B/Ca) to constrain calcifying fluid carbonate chemistry in *P. astreoides* corals that spent their entire life (decades) under acidified low $\Omega_{sw}$ conditions. We found that at the $pH_{cf}$ for corals at the low $\Omega_{sw}$ was slightly lower than at the ambient conditions indicating inability to achieve optimal calcification conditions. We also determined that $pH_{cf}$ and $DIC_{cf}$ are independently regulated and corroborated the calcification response in *P. astreoides* at this site. The study provides new insights into calcification responses of *P. astreoides* under changing environmental conditions and sheds light on the potential of corals to acclimate[30,41,47,61,62]. Using the geochemical proxies in combination with the bio-inorganic model brought forward by

McCulloch et al. [30], we could explain 41% of the variability in coral growth rates along a $\Omega_{sw}$ gradient. The variability which is not explained indicates that additional physiological and environmental processes contribute to the control of calcification rates in natural environments. This provides promising new avenues towards studying acclimation and adaptation potential of long-lived marine invertebrates such as corals.

## Methods

**Site description and coral core collection.** Cores from colonies of *P. astreoides* were collected at the ojos—natural springs of low-pH water—in the National Maritime Park at Puerto Morelos, Mexico (see refs. [11,43] for more details). Five cores were drilled in close proximity to the low pH discharge and seven cores were drilled from control sites outside the ojos discharge influence (~2–5 m away). After collection, cores were dried at 50 °C before further analysis. Water chemistry was measured at the different sites (summarised in Supplementary Table 1 and for more details see refs. [11,33,43,44,63,64]) and used to calculate carbonate chemistry (see Supplementary Table 1). In general, corals were collected from sites that have similar light conditions, differ marginally in temperature (<1 °C lower at the ojos averaged over all seasons with temperatures cooler than ambient in summer and slightly warmer in winter), have consistently lower salinity (2–4 units lower than ambient), and are considerably different in $\Omega_{sw}$ (Supplementary Table 1)[11,44]. We note that these submarine springs are not perfect analogues for future ocean acidification. Specifically, the conditions creating low-pH seawater at the ojos differ from those of the ocean acidification scenario as the high $CO_2$ in the discharging water at the ojos is derived from brackish water that has interacted with soil and limestone. The spring water is characterised by lower pH, higher DIC, higher TA but similar calcium ($Ca^{2+}$) concentration compared to the ambient conditions away from the spring influence. The corals at these ojos are constantly exposed to these discharging water (Supplementary Table 1), as discussed in detail in refs. [11,43,44], and they represent settings with persistent low $\Omega_{sw}$. In particular, because such conditions have persisted at the ojo discharge sites at least since the last deglaciation (~18,000 years ago[65]) the corals at these sites were exposed to low $\Omega_{sw}$ for their whole life span, potentially allowing enough time for acclimation. Moreover, it is quite likely that strong selection processes have resulted in successful colonisation of the ojos by a fraction of the coral population that is better adapted to low $pH_{sw}$ and high $CO_2$.

Water samples were also taken for seawater boron concentrations (measured on a ICP-MS Finnigan Element XR following Krupinski and colleagues[66]; ~430 ± 8 μM, with no difference between ojos and control) and a boron isotopic composition ($\delta^{11}B_{sw}$) of 39.15 (1sd = 0.12; $n$ = 3) for the control site and 38.85 (1sd = 0.17; $n$ = 5) for the low pH ojos. Boron isotopic samples were analysed on a Neptune multi-collector inductively coupled mass spectrometer at National Cheng Kung University, Taiwan, using the standard-sample-standard bracketing technique[67]. The boric acid standard IAEA-B-1 was used as the reference standard (e.g. 39.77‰) to determine the $\delta^{11}B$ of the samples, reproducibility (±0.25‰).

**Sample preparation and geochemical analysis.** Collected coral cores were cut in half. One half was bleached for 24 h, thoroughly washed with milli-Q and dried overnight at 50 °C. Subsequently, the slab surfaces were carefully ground (Struers Silicon carbide grinding paper SiC 500–4000) and briefly polished (Struers DiaPor Dur 9 μm polishing suspension) in preparation for boron analysis using a Struers TegraPol-21 with TegraForce-5 head (Grinder and Polisher). The $\delta^{11}B$ and B/Ca composition was measured simultaneous by laser ablation multi-collector inductively coupled plasma mass spectrometer (Thermo Fisher MC-ICP-MS AXIOM, connected to a UP193fx laser ablation system of New Wave Research, equipped with an excimer 193 nm laser). The measurement procedure followed Fietzke et al.[68] and Wall et al.[40] with slight modifications. Specifically, we used Multiplier and Faraday cups simultaneously to collect data for $B^{10}$ and $B^{11}$ (both on multiplier) as well as $C^{12}$ (Faraday cup). This allows us to derive B/C and $\delta^{11}B$ from the same skeletal material. Similar to previous work the cones were cleaned on a regular basis (every 2–4 days). The tubes going from the ablation cell to the plasma torch were checked for material deposition and cleaned by high flow rates overnight and/or mobilisation of the debris by increased flow rates transporting it out of the tubes. Prior to each measurement session the standard and samples were pre-ablated to remove surface contaminations (spot size used was one size bigger than during analysis). A standard-sample-bracketing method was used. The data of one measurement session contained 5–6 brackets. Both $C^{12}$ and the variation of the standard (NIST SMR610) for each session were used to check for instrument stability and contaminations. Sessions were repeated when the standard drift was higher than the internal reproducibility of the standards (2 SD of the session on the standards). Twenty individual laser tracks (25 × 500 μm) were placed as close as possible to the edge of the skeletal section (expecting to mainly ablate fibres and avoid centres of calcification (COC)), far enough away to avoid ablation through the skeletal part. Yet COC areas may not have been completely avoided. To account for this we: (a) subsequently screened the individual tracks for abnormalities in $C^{12}$ indicative of either ablation through the coral skeletal part (since the underlying skeletal depth is unknown from the surface view, this screening is completed

afterwards) or increased organics and excluded this parts from analysis, and (b) aimed for 20 tracks of ~$25 \times 500 \times 20$ μm on all individuals to have a representative $\delta^{11}B$ dataset per individual. By this approach we expect to cover a representative sample set and minimise the natural intra-skeletal variability and cover similar proportions in each of the different corals (assuming that COC to fibre ratio in coral grown under various environmental conditions stays constant). The accuracy of our $\delta^{11}B$ measurements has been checked by repeated analyses of *Porites* coral standard Jcp-1 and NIST SRM610, measured against a pellet of primary boron standard NBS951 (boric acid) (see Supplementary Fig. 2).

**$\delta^{11}B$ determination**. The data reduction followed Fietzke et al.[68]. This yields one $\delta^{11}B$ value per sample and session with an average precision of <1‰ (1 SD) for ~1.7 μg of carbonate sample. A minimum of 15 and up to 20 values of $\delta^{11}B$ spread over the core surface in the upper few mm of each coral colony (below the tissue, representing ~1 year of growth) were measured to obtain a representative data set per sample. The data set reflects the high variability in $\delta^{11}B$ for a single colony, and replicates were averaged afterwards to yield values that reflect the mean $\delta^{11}B$ value, hence the mean internal calcification conditions (see below).

**B/Ca determination**. B/C elemental ratios have been determined simultaneously with the boron isotope ratios via LA–MC–ICP–MS. Boron isotope data ($^{10}B$ and $^{11}B$) have been collected using a pair of ion counters, while carbon ($^{12}C$) had been determined using a Faraday cup. B/C data are based on the integrated boron intensities ($^{10}B + ^{11}B$) divided by the $^{12}C$ intensity. The calibration (conversion from intensity ratios to concentration ratios) has been done using a natural *L. pertusa* coral sample covering a B concentration range of about 450–950 μmol/mol, which had been determined before using LA–ICP–MS relative to standard NIST-SRM610 using $^{43}Ca$ as internal standard. This calibration procedure resulted in: B/C [μmol/mol] = 78,800 × B/C [cps/cps]; (cps—counts per second, ion beam intensity). We used stoichiometric ratio of C/Ca = 1 as approximation for natural carbonates and translated B/C ratios in B/Ca [μmol/mol] ratios.

**$\delta^{11}B$ as internal pH$_{cf}$ proxy**. All $\delta^{11}B$ values were translated into internal pH$_{cf}$ following Eq. (1) with a seawater $\delta^{11}B_{sw}$ of 38.85 for ojo centres and 39.15 for control sites, a fractionation factor ($\alpha_B$) of 1.0272[69] and p$K^*_B$ averaged for the two sites (see Supplementary Table 1).

$$pH_{cf} = pK_B - \log$$
$$\left[ \left( \delta^{11}B_{sw} - \delta^{11}B \right) / \left( \alpha_B * \delta^{11}B - \delta^{11}B_{SW} + 1000 * (\alpha_B - 1) \right) \right] \quad (1)$$

Following the method in Trotter and colleagues[24] the superimposed physiological pH control was calculated with the equation:

$$\Delta pH = pH_{cf} - pH_{sw} \quad (2)$$

and related to the seawater aragonite saturation state ($\Omega_{sw}$) to quantify the extent of the physiological control on the internal pH$_{cf}$.

We note here, that the local variability in carbonate chemistry at the ojos and hence, associated changes in p$K_B$ and seawater $\delta^{11}B$ can add some uncertainty to the derived pH$_{cf}$ and overestimate or underestimate its actual value. To test the sensitivity to changes in p$K_B$ we used our dataset and recalculated pH$_{cf}$ values. We applied a range of seawater $\delta^{11}B_{sw}$ that encompasses the average measured $\delta^{11}B_{sw}$ per site but also seawater isotopic composition beyond this level ranging from 38.55‰ to 39.45‰ and recalculated pH$_{cf}$ (Supplementary Fig. 3a). This allowed us to decipher the combined role of site specific p$K_B$ and seawater $\delta^{11}B_{sw}$ for a range of skeletal $\delta^{11}B$ (Supplementary Fig. 3b). In general, the $\delta^{11}B$-derived pH$_{cf}$ decreases slightly with increasing seawater $\delta^{11}B$. Changes in seawater $\delta^{11}B_{sw}$ in the corals surrounding will either over or underestimate pH$_{cf}$ and calculated changes in pH$_{cf}$ range from 0.019 to 0.023 pH units per 0.3 change in $\delta^{11}B_{sw}$ (Supplementary Fig. 3c the average difference between our sites; or change from 0.056–0.065 for the entire seawater $\delta^{11}B_{sw}$ range tested). Compared to the pH$_{cf}$ range (8.2–8.8) derived from individually measured skeletal $\delta^{11}B$ values such changes are minor (Supplementary Fig. 3a; in contrast to the individual coral's pH$_{cf}$ standard deviation of 0.04–0.13, Supplementary Table 2).

**B/Ca as CO$_3^{2-}$$_{cf}$ and DIC$_{cf}$ proxy**. All individual B/Ca data were used to estimate CO$_3^{2-}$$_{cf}$ based on the $\delta^{11}B$-derived pH$_{cf}$ data and further used to calculate the DIC$_{cf}$ following the approach of McCulloch et al.[30]. This allows to use the following simplified relationship to determine the CO$_3^{2-}$$_{cf}$ concentration from B/Ca[30]:

$$\left[ CO_3^{2-} \right]_{cf} = \left[ B(OH)_4^- \right]_{cf} * K_D^{B/Ca} / (B/Ca) \quad (3)$$

and the distribution coefficient is determined for synthetic aragonite and follows the equation:

$$K_D^{B/Ca} = 0.00297 \exp\left( -0.0202 [H^+]_{cf} \right) \quad (4)$$

based on the internal pH$_{cf}$[29,30]. Both pH$_{cf}$ and [CO$_3^{2-}$]$_{cf}$ are then used to calculate DIC$_{cf}$.

**Modelling calcification rate using internal pH$_{cf}$ and DIC$_{cf}$**. Calcification rate ($G$) was calculated following McCulloch et al.[19] IpHRAC model:

$$G = k * (\Omega_{cf} - 1)^n \quad (5)$$

The calcification response was calculated with the temperature-dependent rate law constant $k$ and reaction order constant $n$ (applying the equations given in McCulloch et al.[19]: $k = -0.0177*T^2 + 1.47*T + 14.9$ and $n = 0.0628*T + 0.0985$). The individual average temperature data for the different sites were used (Table S1).

For a sole pH-regulation-based model we used seawater DIC concentration DIC$_{sw}$ that were measured at the different sites the individual corals were collected (see Table S1). Aragonite saturation state at the site of calcification ($\Omega_{cf}$) was calculated from pH$_{cf}$ and DIC$_{cf}$ using seacarb. We first followed recent approaches[19,20] by setting DIC$_{cf}$ equivalent to double DIC$_{sw}$[19]. In a second step an advanced bio-inorganic model used both geochemically determined calcification parameters to calculate $\Omega_{cf}$. For the seacarb ($R$) calculations we used the measured average salinity and temperature for the different sites. We assumed [Ca]$^{2+}$ concentrations that equals seawater values to calculate $\Omega_{cf}$.

Both modelled calcification rates were plotted against the measured calcification rates[11], by converting them into relative rates and setting the control site as 1 (or 100%)[19].

**Statistical analysis**. Statistical analysis of the geochemical proxies and derived internal calcification conditions was performed by comparing the two treatment groups (control conditions vs. reduced aragonite saturation state at the centres of the ojos) using Welch's $t$-test (unequal sample numbers). To understand how well the internal calcification conditions ($\Omega_{cf}$—a combined value of both internal pH$_{cf}$ and DIC$_{cf}$) can explain measured changes in net calcification rate we applied simple linear models regressing model-derived relative growth as a function of internal $\Omega_{cf}$. Similarly, changes in geochemical proxies as well as internal conditions were regressed to decipher correlation between these parameters and potential driving forces explaining changes in net calcification rate along the natural environmental seawater $\Omega_{sw}$ gradient[20].

Data analysis and visualisation was done with R Studio version 3.0.1 (R Development Core Team, 2015).

## Data availability

All coral geochemical data and derived calcification conditions are available as Supplementary data file.

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

## Acknowledgements
The research was funded by the German Federal Ministry for Education and Research project BIOACID II (Consortium 3: Natural CO2-rich reefs as windows into the future: Acclimation of marine life to long-term ocean acidification and consequences for bio-geochemical cycle, Grant number: 03F0655A), the Austrian Science Fund Schrödinger Fellowship (funding to M.W., FWF J3667-B25), and National Science Foundation (NSF) OCE-1040952, a University of California Institute for Mexico and the United States (UC-Mexus) grant (to A.P.), and NSF OCE-1041106. E.D.C. was funded through NSF-GFR and a EPA-STAR fellowships. All corals were collected under Secretaría de Agricultura, Ganadería, Desarrollo Rural, Pesca y Alimentación (SAGARPA) Permit DGOPA.00153.170111.-0051 and exported with a Convention on International Trade in Endangered Species (CITES) Permit MX52912.

## Author contribution
M.W., J.F. and A.P. designed the experimental analyses. E.D.C. collected the samples. M.W. prepared the samples. M.W. and J.F. analysed the samples. E.D.C. and A.P. provided background data. M.W. analysed data. M.W., J.F., A.P. and E.D.C. were involved in the preparation of the manuscript.

## Additional information

**Competing interests:** The authors declare no competing interests.

