## [Peer Review File · Nature Communications]

The manuscript, “B isotopes and B/Ca of *Porites astreoides* corals as indicators of calcification processes at low aragonite saturation” investigates the calcifying fluid chemistry of corals growing in naturally low-saturation state seawater. The main findings are that calcifying fluid carbonate ion concentration responds to seawater carbonate chemistry, and this explains some of the calcification response under low saturation state. Although the sample sizes are relatively small, the results provide important new insights into the sensitivity of coral calcification to ocean acidification. In particular, these results from a natural environmental gradient differ from recent results in controlled laboratory experiments, which raises interesting questions for the community to address. I think the data are worthy of publication, but I do have some major suggestions for the manuscript.

Major comments:

- 1) The authors cast their study in large part as a test of the “recently developed bio-inorganic model” of McCulloch et al. (2012). However, in my view, this “model” is already outdated, and by placing too much emphasis on testing this model, the authors are underselling their study. The “model” itself is more of a list of assumptions rather than what is typically meant by the word “model”. It is simply a way to convert pH_{cf} (derived from $\delta^{11}\text{B}$) to Ω_{cf} and calcification rate by assuming that (i) DIC_{cf} is $2X$ DIC_{sw} , (ii) that $[\text{Ca}^{2+}]_{\text{cf}}$ is constant and within a few % of $[\text{Ca}^{2+}]_{\text{sw}}$, and (iii) that bulk coral calcification is a simple function of Ω_{cf} . In 2012, when the “model” was proposed, (i-iii) were not known and the techniques did not exist to test them, and thus this was a reasonable list of assumptions that, at the time, enabled coral calcification rates to be interpreted in relation to $\delta^{11}\text{B}$. However, new techniques have been developed since then, and our understanding of calcifying fluid chemistry has advanced beyond these early assumptions. B/Ca and $\delta^{11}\text{B}$ can now be used in tandem to derive DIC_{cf} and $[\text{CO}_3^{2-}]_{\text{cf}}$. This means that (i) no longer needs to be assumed, which is the most relevant point here because the authors have the B/Ca data to clearly show that assumption (i) is not valid. Additionally, we now know that $[\text{Ca}^{2+}]_{\text{cf}}$ is not constant, but responds to pH_{sw} in some species (DeCarlo et al. 2018a; Comeau et al. 2018). These points do not take away from the authors’ data or their study overall; I am merely saying the authors should go directly to interpreting their $\delta^{11}\text{B}$ and B/Ca data together, rather than restricting themselves to testing an outdated model that uses only $\delta^{11}\text{B}$.
- 2) The authors should go more in-depth in their thinking about DIC_{cf} . Specifically, the authors cast DIC in terms of “ $\text{DIC}_{\text{cf}}/\text{DIC}_{\text{sw}}$ ”, but without justification or a physiological reason why the organism would increase DIC by a specific multiple of the seawater concentration. I understand that this approach has been used before (McCulloch et al. 2012, 2017), but DIC_{sw} was relatively constant in those cases, meaning that those previous authors could have presented their work in terms of absolute DIC_{cf} (or $\text{DIC}_{\text{cf}} = \text{DIC}_{\text{sw}} + X$). The present study is fundamentally different in that DIC_{sw} does vary substantially between the control and treatment locations. Therefore, the present study is unique in that interpretations differ dramatically for absolute DIC_{cf} and $\text{DIC}_{\text{cf}}/\text{DIC}_{\text{sw}}$ (compare Fig. 2C with Fig. S1). I actually see this as an opportunity for the authors to test some existing ideas about DIC_{cf} and its relation to seawater chemistry, so this is more of a suggestion to incorporate some additional interesting aspects to the manuscript, rather than a criticism. For instance, Comeau et al. (2018) found strong relationships between DIC_{cf} and DIC_{sw} in laboratory

experiments, which obviously differs from the present findings, but sets up an insightful comparison of laboratory and field results.

- 3) It should be described clearly that the information provided by $\delta^{11}\text{B}$ and B/Ca is $[\text{CO}_3^{2-}]_{\text{cf}}$, *not* Ω_{cf} . The two differ depending on $[\text{Ca}^{2+}]_{\text{cf}}$, which the authors have no data to constrain. In my view, the text and figure labels should all be “ $[\text{CO}_3^{2-}]_{\text{cf}}$ ”, and not “ Ω_{cf} ”. If the authors really wish to use “ Ω_{cf} ”, then it needs to be clearly explained in the main text and figure captions that Ω_{cf} can only be estimated here by making an assumption about $[\text{Ca}^{2+}]_{\text{cf}}$.
- 4) There is a substantial amount of very recent literature that is not cited, but that is highly relevant to this study (DeCarlo et al. 2017, 2018b, a, Ross et al. 2018b, a; Comeau et al. 2018; Cornwall et al. 2018). I recognize that these papers were published in the past year, and that they are all from our group. Thus, I am not trying to push our own interpretations or force the authors to cite our work, but rather I want to make sure the authors are aware of these papers because they represent recent applications of $\delta^{11}\text{B}$ and B/Ca in tandem, new techniques to quantify Ω_{cf} and $[\text{Ca}^{2+}]_{\text{cf}}$, insights into controls on calcifying fluid chemistry, other field data to compare with the present study, and codes for computing systematic and non-systematic uncertainties associated with the combined $\delta^{11}\text{B}$ and B/Ca proxy approach. It is at the authors’ discretion if, and how, to integrate these references into their manuscript.
- 5) Methods: some clarification of the proxy calculations is needed. Interpretations of the boron-based proxies are sensitive to salinity, temperature, $\delta^{11}\text{B}_{\text{sw}}$, and $[\text{B}]_{\text{sw}}$ (see DeCarlo et al. 2018b). Since some of these parameters vary between ojos and control sites, careful consideration is required. I commend the authors for measuring seawater $[\text{B}]$ and $\delta^{11}\text{B}$, as these could depart from typical values within the ojos. However, more information is needed. How many water samples were taken and at how many different times? The seawater $[\text{B}]$ is presented as “ $\sim 430 \mu\text{M}$ ”, but how much did it vary? The seawater $\delta^{11}\text{B}$ was different between the ojos and control sites, which must influence the pH_{cf} interpretations. How influential is this difference in seawater $\delta^{11}\text{B}$ (i.e. how much would results change if $\delta^{11}\text{B}$ was assumed constant)? Since the seawater $[\text{B}]$ and $\delta^{11}\text{B}$ could change over time (and this is presumably not captured by the water sampling), some assessment of their potential influence on the results is needed. Another potential issue is that the authors used a constant pK_{B} (averaged from the two sites) for the pH_{cf} calculations, but site-specific pK_{B} for the B/Ca calculations. I do not understand this approach, and I wonder how much it influences the results.

Minor comments:

- 1) I suggest revising the title. At present, it sounds as if the study develops a new technique to study coral calcification based on boron isotopes and B/Ca. But those techniques have already been developed. This study applies those techniques to an interesting natural gradient in seawater chemistry.
- 2) Abstract: The statement “were found growing in natural settings where the aragonite saturation state is very low” should be revised because this study did not find these corals. The corals and the setting were described by some of these authors previously;

the present study applies geochemical proxies to the previously-described corals.

- 3) Abstract: “decades” - is evidence presented here that these corals have been alive for multiple decades?
- 4) Introduction: I suggest not to cite IPCC for ocean acidification affecting coral calcification. There are many primary articles to support this statement.
- 5) Second sentence of the introduction: Ref 15 does not show calcification changes, as is implied. Barkley et al (2015) is the appropriate reference showing calcification changes at this site.
- 6) Introduction: “Number of tropical corals (40%)”? Is this species or something else?
- 7) “Most studies suggest that seawater pH_{sw} is the main driver affecting pH_{cf} ”: References to support this statement?
- 8) I do not think it is appropriate to say, “slightly different salinity”. Even 1-3 PSU differences can be meaningful. Just report the actual differences.
- 9) It would be more appropriate to plot DIC_{cf} in the main text. As the authors note, $\text{DIC}_{\text{cf}}/\text{DIC}_{\text{sw}}$ changes mainly due to DIC_{sw} , not DIC_{cf} . Previous studies have interpreted $\text{DIC}_{\text{cf}}/\text{DIC}_{\text{sw}}$ as indicative of calcification processes because in those studies DIC_{sw} was relatively constant. Conversely, in the present study, $\text{DIC}_{\text{cf}}/\text{DIC}_{\text{sw}}$ really just tells us about the environment, not differences in the calcification process between treatment and control corals.
- 10) Discussion: “This points towards additional controlling mechanisms...” should also mention $[\text{Ca}]_{\text{cf}}$ here because what the authors call “ Ω_{cf} ” is really just $[\text{CO}_3^{2-}]_{\text{cf}}$, which can depart from Ω_{cf} depending on $[\text{Ca}]_{\text{cf}}$. This is especially relevant because, at least in some species, $[\text{Ca}]_{\text{cf}}$ depends on pH_{sw} (DeCarlo et al. 2018a; Comeau et al. 2018).
- 11) “The implementation and application of a combined $\delta^{11}\text{B-B/Ca}$, with B/Ca serving as a proxy for CO_3^{2-} concentration and, subsequently, DIC_{cf} is a relatively new approach²⁹” I think reference 29 is a typo here because that study did not use B/Ca to determine $[\text{CO}_3^{2-}]$ or DIC.
- 12) Need to be consistent with “acclimation” and “acclimatization”.
- 13) Methods: “First seawater boron concentration was determined based on the site-specific pK_{B} assuming boron concentration at the site of calcification equals to seawater boron concentration.” I assume the authors mean “First, seawater *borate*...”.

References cited:

- Barkley HC, Cohen AL, Golbuu Y, Starczak VR, DeCarlo TM, Shamberger KE (2015) Changes in coral reef communities across a natural gradient in seawater pH. *Sci Adv* 1:e1500328.
- Comeau S, Cornwall CE, DeCarlo TM, Krieger E, McCulloch MT (2018) Similar controls on

- calcification under ocean acidification across unrelated coral reef taxa. *Glob Chang Biol* 24:4857–4868. doi: 10.1111/gcb.14379
- Cornwall CE, Comeau S, DeCarlo TM, Moore B, D’Alexis Q, McCulloch MT (2018) Resistance of corals and coralline algae to ocean acidification: physiological control of calcification under natural pH variability. *Proc R Soc B Biol Sci* 285:20181168. doi: 10.1098/rspb.2018.1168
- DeCarlo TM, D’Olivo JP, Foster T, Holcomb M, Becker T, McCulloch MT (2017) Coral calcifying fluid aragonite saturation states derived from Raman spectroscopy. *Biogeosciences* 14:5253–5269. doi: 10.5194/bg-14-5253-2017
- DeCarlo TM, Comeau S, Cornwall CE, McCulloch MT (2018a) Coral resistance to ocean acidification linked to increased calcium at the site of calcification. *Proc R Soc B Biol Sci* 285:20180564. doi: 10.1098/rspb.2018.0564
- DeCarlo TM, Holcomb M, McCulloch MT (2018b) Reviews and syntheses: Revisiting the boron systematics of aragonite and their application to coral calcification. *Biogeosciences Discuss* 1–20. doi: 10.5194/bg-2018-77
- McCulloch MT, Falter J, Trotter J, Montagna P (2012) Coral resilience to ocean acidification and global warming through pH up-regulation. *Nat Clim Chang* 2:623–627.
- McCulloch MT, D’Olivo JP, Falter J, Holcomb M, Trotter JA (2017) Coral calcification in a changing World and the interactive dynamics of pH and DIC upregulation. *Nat Commun* 8:15686. doi: 10.1038/ncomms15686
- Ross CL, DeCarlo TM, McCulloch MT (2018a) Environmental and physiochemical controls on coral calcification along a latitudinal temperature gradient in Western Australia. *Glob Chang Biol*. doi: 10.1111/gcb.14488
- Ross CL, Schoepf V, DeCarlo TM, McCulloch MT (2018b) Mechanisms and seasonal drivers of calcification in the temperate coral *Turbinaria reniformis* at its latitudinal limits. *Proc R Soc B Biol Sci* 285:20180215. doi: 10.1098/rspb.2018.0215

03 December 2018

- Thomas M. DeCarlo

Reviewer #2 (Remarks to the Author):

Wall et al., have utilized $\delta^{11}\text{B}$ and B/Ca measurements in *Porites astreoides* in order to derive the calcifying pH_{cf}, DIC_{cf} and subsequent Ω_{cf} . The idea was to understand the acclimation of corals in natural low Ω_{sw} environment from species collected in the NMP at Puerto Morelos. In order to achieve that, they used 2 close locations with different environmental settings and Ω_{sw} . Corals from both locations exhibit up-regulated pH_{cf}, DIC_{cf} with more up-regulation at the low pH centers, however, Ω_{cf} for both locations differed and the low pH locations presented a lower Ω_{cf} compared to control. This difference is consistent with the lower calcification (~37%) observed at the low pH locations but this lower calcification is not fully explained by the decrease in Ω_{cf} (model 44%).

Due to the growing interest of the boron based proxies in corals, this paper adds a stone to the growing evidence of a dynamic calcifying fluid (pH_{cf}, DIC_{cf}) in natural environments modulating the calcification response. This paper highlights the need for a better understanding of the symbiotic relationship of corals under different environments.

In terms of analysis, it would be great to have some standards presented in the paper to give more confidence to the data, even if they look consistent.

Minor points:

Line 26: It is a bit confusing here. "whether these corals", may be change for "in what extent these corals have acclimated".

Line 28: "increase the pH" may be change for "increase the pH and subsequent Ω_{cf} " since Ω_{cf} is likely the main trigger for calcification.

Line 47: May be add Ries et al., 2009.

Line 54: May be add "and upregulate their Ω_{cf} ".

Line 91: They do not have similar temperature or salinity, 1-2 °C lower and 3psu lower in the "low pH centers" compared to centers. Do you have an idea of the seasonal variability? Can you test whether those differences could explain a part of your variability?

Line 98: Could the acclimation come from this high Ca content as well?

Line 181: This result is central in your paper might be good to explain a bit more at line 439.

Line 197: “thereby modulating Ω_{cf} and calcification rate”

Line 210: May be add a line saying that the temperature is not the main driver because people will look for this relationship, and that will reinforce your arguments.

At “low pH sites” the temperature is lower than at control sites, since a decrease in pH_{cf} and DIC_{cf} is observed with increasing temperature, temperature in our case can't explain the decrease of pH_{cf} and DIC_{cf}.

Line 217: I think this paragraph needs to move or towards the conclusion or in the introduction.

Line 262: Is there any data for your corals?

Line 265: So Ω_{cf} explains 44% of the calcification rate variability, all the arguments here are likely to modulate the Ω_{cf} but since it is only 44%, any ideas what are the independent (from the calcifying fluid) triggers for the 56% left?

Line 284: Can you explain why it is controlled by the adjacent epithelium?

Line 290: May be add a sentence to conclude more smoothly.

Line 320: How did you measure your seawater samples?

Line 334: What are those modifications?

Line 360: Is it due to the natural variability or to the method? Do you have any standards ran in liquid during your session?

Line 367: Explain quickly B/C, it is confusing with B/Ca even if well explained in the rest of the paragraph. Have you compared your B/Ca with liquid measurements?

Line 439: May be reformulate.

Table S2: 1st column, typo 2 decimals

Reviewer #1

Many of the comments of reviewer #1 are related hence we grouped these comments and responded to them together. We then respond to additional comments separately.

The manuscript, “B isotopes and B/Ca of *Porites astreoides* corals as indicators of calcification processes at low aragonite saturation” investigates the calcifying fluid chemistry of corals growing in naturally low-saturation state seawater. The main findings are that calcifying fluid carbonate ion concentration responds to seawater carbonate chemistry, and this explains some of the calcification response under low saturation state. Although the sample sizes are relatively small, the results provide important new insights into the sensitivity of coral calcification to ocean acidification. In particular, these results from a natural environmental gradient differ from recent results in controlled laboratory experiments, which raises interesting questions for the community to address. I think the data are worthy of publication, but I do have some major suggestions for the manuscript.

Major comments:

1) The authors cast their study in large part as a test of the “recently developed bioinorganic model” of McCulloch et al. (2012). However, in my view, this “model” is already outdated, and by placing too much emphasis on testing this model, the authors are underselling their study. The “model” itself is more of a list of assumptions rather than what is typically meant by the word “model”. It is simply a way to convert pH_{cf} (derived from $\delta_{11}\text{B}$) to Ω_{cf} and calcification rate by assuming that (i) DIC_{cf} is 2X DIC_{sw} , (ii) that $[\text{Ca}^{2+}]_{\text{cf}}$ is constant and within a few % of $[\text{Ca}^{2+}]_{\text{sw}}$, and (iii) that bulk coral calcification is a simple function of Ω_{cf} . In 2012, when the “model” was proposed, (i-iii) were not known and the techniques did not exist to test them, and thus this was a reasonable list of assumptions that, at the time, enabled coral calcification rates to be interpreted in relation to $\delta_{11}\text{B}$. However, new techniques have been developed since then, and our understanding of calcifying fluid chemistry has advanced beyond these early assumptions. B/Ca and $\delta_{11}\text{B}$ can now be used in tandem to derive DIC_{cf} and $[\text{CO}_3^{2-}]_{\text{cf}}$. This means that (i) no longer needs to be assumed, which is the most relevant point here because the authors have the B/Ca data to clearly show that assumption (i) is not valid. Additionally, we now know that $[\text{Ca}^{2+}]_{\text{cf}}$ is not constant, but responds to pH_{sw} in some species (DeCarlo et al. 2018a; Comeau et al. 2018). These points do not take away from the authors’ data or their study overall; I am merely saying the authors should go directly to interpreting their $\delta_{11}\text{B}$ and B/Ca data together, rather than restricting themselves to testing an outdated model that uses only $\delta_{11}\text{B}$.

2) The authors should go more in-depth in their thinking about DIC_{cf} . Specifically, the authors cast DIC in terms of “ $\text{DIC}_{\text{cf}}/\text{DIC}_{\text{sw}}$ ”, but without justification or a physiological reason why the organism would increase DIC by a specific multiple of the seawater concentration. I understand that this approach has been used before (McCulloch et al. 2012, 2017), but DIC_{sw} was relatively constant in those cases, meaning that those previous authors could have presented their work in terms of absolute DIC_{cf} (or $\text{DIC}_{\text{cf}} = \text{DIC}_{\text{sw}} + X$). The present study is fundamentally different in that DIC_{sw} does vary substantially between the control and treatment locations. Therefore, the present study is unique in that interpretations differ dramatically for absolute DIC_{cf} and $\text{DIC}_{\text{cf}}/\text{DIC}_{\text{sw}}$ (compare Fig. 2C with Fig. S1). I actually see this as an opportunity for the authors to test some existing ideas about DIC_{cf} and its relation to seawater chemistry, so this

is more of a suggestion to incorporate some additional interesting aspects to the manuscript, rather than a criticism. For instance, Comeau et al. (2018) found strong relationships between DIC_{cf} and DIC_{sw} in laboratory experiments, which obviously differs from the present findings, but sets up an insightful comparison of laboratory and field results.

3) It should be described clearly that the information provided by $\delta_{11}B$ and B/Ca is $[CO_{32-}]_{cf}$, *not* Ω_{cf} . The two differ depending on $[Ca_{2+}]_{cf}$, which the authors have no data to constrain. In my view, the text and figure labels should all be “[$CO_{32-}]_{cf}$ ”, and not “ Ω_{cf} ”. If the authors really wish to use “ Ω_{cf} ”, then it needs to be clearly explained in the main text and figure captions that Ω_{cf} can only be estimated here by making an assumption about $[Ca_{2+}]_{cf}$.

4) There is a substantial amount of very recent literature that is not cited, but that is highly relevant to this study (DeCarlo et al. 2017, 2018b, a, Ross et al. 2018b, a; Comeau et al. 2018; Cornwall et al. 2018). I recognize that these papers were published in the past year, and that they are all from our group. Thus, I am not trying to push our own interpretations or force the authors to cite our work, but rather I want to make sure the authors are aware of these papers because they represent recent applications of $\delta_{11}B$ and B/Ca in tandem, new techniques to quantify Ω_{cf} and $[Ca_{2+}]_{cf}$, insights into controls on calcifying fluid chemistry, other field data to compare with the present study, and codes for computing systematic and non-systematic uncertainties associated with the combined $\delta_{11}B$ and B/Ca proxy approach. It is at the authors’ discretion if, and how, to integrate these references into their manuscript.

And Minor points:

10) Discussion: “This points towards additional controlling mechanisms...” should also mention $[Ca]_{cf}$ here because what the authors call “ Ω_{cf} ” is really just $[CO_{32-}]_{cf}$, which can depart from Ω_{cf} depending on $[Ca]_{cf}$. This is especially relevant because, at least in some species, $[Ca]_{cf}$ depends on pH_{sw} (DeCarlo et al. 2018a; Comeau et al. 2018).

9) It would be more appropriate to plot DIC_{cf} in the main text. As the authors note, DIC_{cf}/DIC_{sw} changes mainly due to DIC_{sw} , not DIC_{cf} . Previous studies have interpreted DIC_{cf}/DIC_{sw} as indicative of calcification processes because in those studies DIC_{sw} was relatively constant. Conversely, in the present study, DIC_{cf}/DIC_{sw} really just tells us about the environment, not differences in the calcification process between treatment and control corals.

We thank the reviewer for recognizing the importance and relevance of this paper and in particular, for his valuable comments that helped to improve and better structure our manuscript. We also thank for referring us to the recently published very relevant papers in his comments #1-4, 10, 9. These references were not used in the original manuscript primarily because the manuscript has been written before most of these papers were published and has been under review for a while. Yet we certainly agree that this are very relevant aspects and we are including these important concepts, mechanisms and references in the revised version.

First of all, in the revised version we now focused only on the model where both $\delta^{11}B$ and B/Ca are applied together. In the results section we now refer just to this newer approach (Line 162: “We used the combined pH_{cf} and the B/Ca-derived DIC_{cf} concentration at the site of calcification for both ojo and control corals.”). We also adjusted the discussion by only talking about the ability to use our dual-

geochemical proxy data to elucidate underlying mechanisms and potential drivers. We updated the introduction and when we talk about the model we used the opening of this paragraph to state (Line 244): “The application of our dual geochemical proxy data to model coral growth (e.g. IpHRAC^{21,33}) allowed us to further pinpoint potential mechanisms of how external seawater conditions affect internal calcifying conditions and ultimately skeletal growth.” We used this to directly enter the discussion on DIC_{sw} variations and their role in modulating or not DIC_{cf} (more to this topic see below). We moved the old model to the supplements (Fig. S1) because it is used to support the point at the beginning of our discussion (Line 190): “Interestingly at this field site ojos with low aragonite saturation state had elevated DIC_{sw}, but this did not result in higher DIC_{cf} concentrations in the calcifying fluids, indicating a decoupling of internal and external DIC concentrations.”, and used this opportunity to further emphasize this point. This also is in-line with the comment on DIC_{cf} and we specifically addressed this issue now in more detail in Line 251: “...At our study sites, DIC_{sw} significantly and extensively differs between sites, with higher values at the low Ω_{sw} sites (in average 2790 $\mu\text{mol kg}^{-1}$ compared to control average DIC_{sw} of 2050 $\mu\text{mol kg}^{-1}$). ... If corals modify internal DIC_{cf} by simply up-regulating DIC_{cf} from the external concentrations baseline, we would expect higher DIC_{cf} values for the ojo corals where DIC_{sw} is higher. Under such assumption the elevated DIC_{cf} compensates for the slightly lower pH_{cf} effect on Ω_{cf} and calcification rates would essentially be similar between sites (Fig. S1).”

We also adjusted Figure 2 by adding the supplementary figure of DIC_{sw} here, so it already clearly visualizes that DIC_{cf}/DIC_{sw} observed response to seawater Ω_{sw} is mainly due to the substantial and significant change in seawater DIC_{sw} pinpointing towards the above mentioned decoupling of seawater DIC and internal DIC_{cf}. We added this to the discussion (Line 259): “However our data clearly demonstrates that DIC_{cf} is not directly linked to external concentrations and can differ significantly from that of seawater^{25,33,52} (reported DIC_{cf}-upregulation values range from 1.6-3.2^{33,44}, with the ojos corals at our sites are below the lower end), and this impacts Ω_{cf} (more precisely CO_3^{2-}) and calcification. A recent study under laboratory conditions with *Stylophora pistillata*⁵³ observed that changes in DIC_{sw} concentration modulates internal pH_{cf} regulation, with higher external DIC_{cf} facilitating higher internal pH_{cf} observing a clear correlation between seawater DIC_{sw}/H⁺_{sw} and pH_{cf}. Since DIC_{sw} at the ojos is significantly higher than at the control sites, one could expect this compensates reduced pH_{cf} up-regulation induced only due to changes in seawater pH_{sw}. Yet we do not see a strong correlation between pH_{cf} and seawater DIC_{sw}/H⁺_{sw}, suggesting different drivers for pH_{cf} regulation in *Porites astreoides* compared to those observed in *Stylophora pistillata*⁵³. Nevertheless, the change in pH_{cf} and the limited ability to upregulate DIC_{cf} beyond seawater DIC_{sw} at the ojos corroborates the observed calcification rate decrease at the ojos.”

Regarding comment #3 we agree that we need to be more precise here and clearly state that our Ω_{cf} is based on the two derived carbonate chemistry parameters from our proxies and not based on the more relevant Ω_{cf} derived from CO_3^{2-} and Ca^{2+}_{cf} . Plotting the data against CO_3^{2-} only, would not show the observed relationship since our conclusion is that the COMBINED even though limited change in pH_{cf} and CO_3^{2-} (expressed as Ω_{cf}) concentration along the Ω_{sw} gradient contributes to the change in calcification rate. We want to specifically make this point for reasons mentioned by the reviewer but also to point out that we need to be cautious with relationships that may not be significant yet in combination with each other can provide some explanation of/new insights to observed patterns. This relationship pinpoints towards the points raised in the reviewer’s comment #4 that Ca^{2+}_{cf} may be changed along the Ω_{sw} gradient and likely drives the observed decline in calcification rate. In the revised version we specifically address this issue and we thank the reviewer for encouraging us to take the discussion into this direction: eg. We now state in Line (275): “Recent studies identified internal calcium (Ca^{2+}_{cf}) regulation as an additional player in coral calcification responses and emphasized that

regulation of $\text{Ca}^{2+}_{\text{cf}}$ can contribute to a corals' resistance to future ocean changes^{42-44,54}. In this sense, the good agreement of our model with the observed calcification response potentially indicates that internal average steady-state calcium concentrations ($\text{Ca}^{2+}_{\text{cf}}$) are lower at the ojos by some proportion that is related to the pH_{cf} changes, since our model based on pH_{cf} and $\text{CO}_3^{2-}_{\text{cf}}$ can reproduce the observed calcification decline. This suggests a strong link between $\text{Ca}^{2+}_{\text{cf}}$ and pH_{cf} and supports the idea of a plasma-membrane Ca-ATPase^{55, but see 56} responsible for pH_{cf} regulation. However, it is possible that pH_{cf} but also $\text{Ca}^{2+}_{\text{cf}}$ was regulated by additional and/or different ion transport mechanisms (e.g. potentially ion exchangers, Ca^{2+} -channels)^{56,57}. ” and in (Line 288): “The present study also indicates that the acclimation process in different corals encompassed some degree of flexibility in terms of the relative role of pH_{cf} and DIC_{cf} regulation in increasing the Ω_{cf} , with some individuals compensating by adjusting their internal pH_{cf} and others primarily by DIC_{cf} modulation. This may also be true of the role of $\text{Ca}^{2+}_{\text{cf}}$ upregulation. The relative amount, source and transportation pathways of DIC , H^+ and Ca^{2+} to the site of calcification are still not fully understood⁵⁸ and transport processes may differ between individual corals.”

5) Methods: some clarification of the proxy calculations is needed. Interpretations of the boron-based proxies are sensitive to salinity, temperature, $\delta_{11}\text{B}_{\text{sw}}$, and $[\text{B}]_{\text{sw}}$ (see DeCarlo et al. 2018b). Since some of these parameters vary between ojos and control sites, careful consideration is required. I commend the authors for measuring seawater $[\text{B}]$ and $\delta_{11}\text{B}$, as these could depart from typical values within the ojos. However, more information is needed. How many water samples were taken and at how many different times? The seawater $[\text{B}]$ is presented as “~430 [M”, but how much did it vary? The seawater $\delta_{11}\text{B}$ was different between the ojos and control sites, which must influence the pH_{cf} interpretations. How influential is this difference in seawater $\delta_{11}\text{B}$ (i.e. how much would results change if $\delta_{11}\text{B}$ was assumed constant)? Since the seawater $[\text{B}]$ and $\delta_{11}\text{B}$ could change over time (and this is presumably not captured by the water sampling), some assessment of their potential influence on the results is needed.

All water samples (in total 8) were first analyzed on ICP-MS Finnigan Element XR for boron concentration measurements following Krupinski et al. (2017) before they were further analyzed on a Neptune multi-collector ICP-MS at National Cheng Kung University, Taiwan for seawater $\delta^{11}\text{B}$. Within these set of water samples (n=8 for both control and ojo; 3 ojos and 5 control) the variation was rather low (2%) and agree with typical open ocean B concentrations reported ($[\text{B}] = 432.9 \mu\text{mol kg}^{-1} \pm 2$; (Lee et al., 2010)). Since also no differences could be found between sites it did not require adjustments in terms of boron concentration in our calculations. Our seawater $\delta^{11}\text{B}$ composition, though, indicate minor differences in composition between the different sites investigated in this study. Hence, we used our dataset to illustrate the relationships of changing seawater $\delta^{11}\text{B}$ and pH_{cf} when we left the different pK_B for each site constant. We applied a range of $\delta^{11}\text{B}$ that encompasses the average measured $\delta^{11}\text{B}$ per site but also seawater isotopic composition beyond this level ranging from 38.55 to 39.45 ‰ and recalculated pH_{cf} from the entire dataset (Fig. 1A). This allowed us to decipher the combined role of site specific pK_B and seawater $\delta^{11}\text{B}$ for a range of skeletal $\delta^{11}\text{B}$ (Fig. 1B). In general, the $\delta^{11}\text{B}$ derived pH_{cf} decreases with increasing seawater $\delta^{11}\text{B}$. Changes in seawater $\delta^{11}\text{B}$ in the corals surrounding will either over or underestimate pH_{cf} and calculated changes in pH_{cf} are with 0.019 to 0.023 pH units per 0.3 change in $\delta^{11}\text{B}_{\text{sw}}$ (Fig. 1C; the average difference between our sites; or 0.056-0.065 for the entire $\delta^{11}\text{B}_{\text{sw}}$ range tested) in the range observed pH_{cf} for the investigated corals 8.88-8.18 (Fig. 1A; $\Delta\text{pH}_{\text{cf}} = 0.8$; or sd per individual ranging from: 0.04-0.13). Of course such changes add some uncertainty and potentially also contribute to the fact that our model only explains 44% of the

observed variation. We added this into the Method section in Line 428: “We note here, that the local variability in carbonate chemistry at the ojos and hence, associated changes in pK_B and seawater $\delta^{11}B$ can add some uncertainty to the derived pH_{cf} and over- or underestimate its actual value. To test the sensitivity to changes in pK_B we used our dataset and recalculated pH_{cf} values. We applied a range of seawater $\delta^{11}B_{sw}$ that encompasses the average measured $\delta^{11}B_{sw}$ per site but also seawater isotopic composition beyond this level ranging from 38.55 to 39.45 ‰ and recalculated pH_{cf} (supplementary figure S3A). This allowed us to decipher the combined role of site specific pK_B and seawater $\delta^{11}B_{sw}$ for a range of skeletal $\delta^{11}B$ (supplementary figure. S3B). In general, the $\delta^{11}B$ derived pH_{cf} decreases slightly with increasing seawater $\delta^{11}B$. Changes in seawater $\delta^{11}B_{sw}$ in the corals surrounding will either over or underestimate pH_{cf} and calculated changes in pH_{cf} range from 0.019 to 0.023 pH units per 0.3 change in $\delta^{11}B_{sw}$ (supplementary figure S3C the average difference between our sites; or change from 0.056-0.065 for the entire seawater $\delta^{11}B_{sw}$ range tested). Compared to the pH_{cf} range (8.2-8.8) derived from individually measured skeletal $\delta^{11}B$ values such changes are minor (Fig. 1A; $\Delta pH_{cf} = 0.8$; in contrast to the individual coral’s pH_{cf} standard deviation of 0.04-0.13, supplementary table S2).”

Fig. S3: pH_{cf} systematics as a function of changes in seawater $\delta^{11}B_{sw}$. A) pH_{cf} for the entire skeletal $\delta^{11}B$ for different seawater $\delta^{11}B_{sw}$. B) pH_{cf} as a function of aragonite $\delta^{11}B$ calculated from different $\delta^{11}B_{sw}$ conditions and C) how a change in seawater $\delta^{11}B_{sw}$ changes pH_{cf} (ΔpH_{cf}) over the entire skeletal $\delta^{11}B$ range.

Another potential issue is that the authors used a constant pK_B (averaged from the two sites) for the pH_{cf} calculations, but site-specific pK_B for the B/Ca calculations. I do not understand this approach, and I wonder how much it influences the results.

We apologize but this was actually an error in the manuscript. We used different pK_B for both pH_{cf} and B/Ca calculations.

Minor comments:

1) I suggest revising the title. At present, it sounds as if the study develops a new technique to study coral calcification based on boron isotopes and B/Ca. But those techniques have already been developed. This study applies those techniques to an interesting natural gradient in seawater chemistry.

We changed the title to: Using B isotopes and B/Ca in corals from low saturation springs to constrain calcification mechanisms

2) Abstract: The statement “were found growing in natural settings where the aragonite saturation state is very low” should be revised because this study did not find these corals. The corals and the setting were described by some of these authors previously;

We changed this to (Line 22): “Here we measured geochemical proxies ($\delta^{11}B$ and B/Ca) of these previously described *Porites astreoides* corals that have been growing for their entire life (decades) under conditions of low seawater aragonite saturation (Ω_{sw} : 0.77-1.85)¹,..”

3) Abstract: “decades” - is evidence presented here that these corals have been alive for multiple decades?

Yes, based on annual bands analyzed by Crook et al 2013 we have some that are at least 30 years old.

4) Introduction: I suggest not to cite IPCC for ocean acidification affecting coral calcification. There are many primary articles to support this statement.

We changed the citation.

5) Second sentence of the introduction: Ref 15 does not show calcification changes, as is implied. Barkley et al (2015) is the appropriate reference showing calcification changes at this site.

Reference changed

6) Introduction: “Number of tropical corals (40%)”? Is this species or something else?

The referred meta-analysis by Chan and Connell used different studies (fulling their criteria that studied different but also the same species) and treated all individually (mixing species, sites, etc). In the revised version we deleted the percentage to not imply that it refers to 40% of all tropical coral species or anything else. We just state that (Line 45): “These efforts have provided strong evidence that the calcification rates of a large number of coral species investigated to date will decline in response to projected $p\text{CO}_2^{17}$.”

7) “Most studies suggest that seawater pH_{sw} is the main driver affecting pH_{cf} ”: References to support this statement?

We adjusted the text here and added references studying the relationship between internal and external seawater pH.

8) I do not think it is appropriate to say, “slightly different salinity”. Even 1-3 PSU differences can be meaningful. Just report the actual differences.

Change to (Line 96): “.. and have consistently lower salinity (2-4 psu lower than ambient), ...”

11) “The implementation and application of a combined $\delta_{11}\text{B-B/Ca}$, with B/Ca serving as a proxy for CO_3^{2-} concentration and, subsequently, DIC_{cf} is a relatively new approach 29” I think reference 29 is a typo here because that study did not use B/Ca to determine $[\text{CO}_3^{2-}]$ or DIC.

Correct the reference and add the new references

12) Need to be consistent with “acclimation” and “acclimatization”.

Changed

13) Methods: “First seawater boron concentration was determined based on the site specific pK_{B} assuming boron concentration at the site of calcification equals to seawater boron concentration.” I assume the authors mean “First, seawater *borate*...”.

Yes that is correct, we now just referred to the publication where the procedure and all the details are provided.

Reviewer #2 (Remarks to the Author):

Wall et al., have utilized $\delta_{11}\text{B}$ and B/Ca measurements in *Porites astreoides* in order to derive the

calcifying pHcf, DICcf and subsequent Ω_{cf} . The idea was to understand the acclimation of corals in natural low Ω_{sw} environment from species collected in the NMP at Puerto Morelos. In order to achieve that, they used 2 close locations with different environmental settings and Ω_{sw} . Corals from both locations exhibit up-regulated pHcf, DICcf with more up-regulation at the low pH centers, however, Ω_{cf} for both locations differed and the low pH locations presented a lower Ω_{cf} compared to control. This difference is consistent with the lower calcification (~37%) observed at the low pH locations but this lower calcification is not fully explained by the decrease in Ω_{cf} (model 44%). Due to the growing interest of the boron based proxies in corals, this paper adds a stone to the growing evidence of a dynamic calcifying fluid (pHcf, DICcf) in natural environments modulating the calcification response. This paper highlights the need for a better understanding of the symbiotic relationship of corals under different environments.

We thank the reviewer for an accurate summary of our work and major findings and for recognizing the importance and relevance of this paper.

1. In terms of analysis, it would be great to have some standards presented in the paper to give more confidence to the data, even if they look consistent.

A figure showing the analyses of standards is added to the supplemental material (Fig. S2).

Fig. S2: Repeated LA-MC-ICPMS analyses of pellets of coral standard Jcp-1 and boric acid standard NBS951. Individual runs consisted of 40s background and 60s ablation data collection. All individual run data have been normalized to the mean of all NBS951 (0.0 ± 0.4 ‰; 2se) data to allow for testing the reproducibility of both standards' measurements (Jcp-1: 24.0 ± 0.4 ‰; 2se). Note: no drift correction has been applied. Repeated measurements of glass standard NIST610 vs. NBS951 pellet yielded a mean $\delta^{11}\text{B}$ for NIST610 of 0.3 ± 0.2 ‰ (2se, n=160).

Minor points:

2. Line 26: It is a bit confusing here. "whether these corals", may be change for "in what extent these corals have acclimated".

This change has been made in the revised version of the manuscript.

3. Line 28: “increase the pH” may be change for “increase the pH and subsequent Ω_{cf} ” since Ω_{cf} is likely the main trigger for calcification.

We changed it to (Line 26): “...addressing their ability to manipulate the carbonate chemistry of the calcifying fluid, including the pH (pH_{cf}), dissolved inorganic carbon (DIC_{cf}) concentration, and subsequently carbonate ion concentration ($CO_3^{2-}_{cf}$).” We used CO_3^{2-} instead of Ω_{cf} considering the first reviewers comment that more precisely with our method we characterized $CO_3^{2-}_{cf}$ and not Ω_{cf} .

4. Line 47: May be add Ries et al., 2009.

We added the reference when we talk about (Line 47): “... that certain coral species were able to maintain high calcification rates or even benefit from elevated $pCO_2^{2,18-20}$ suggesting ...”.

5. Line 54: May be add “and upregulate their Ω_{cf} ”.

The paragraph focuses on pH_{cf} upregulation capacity of corals and the $d^{11}B$ proxy and not the potential consequences such as changes in Ω_{cf} . However, only a few sentence further below (now Line 65) when we start addressing the corals ability to up-regulate DIC_{cf} we discuss the change in Ω_{cf} . We adjusted this sentence to clarify that the upregulation includes both pH and DIC (Line 69): “Together – the potential to upregulate DIC_{cf} and pH_{cf} - allows for higher carbonate ion concentrations at the site of calcification and hence a higher Ω_{cf} that facilitates calcification^{32,23}.”

6. Line 91: They do not have similar temperature or salinity, 1-2 °C lower and 3psu lower in the “low pH centers” compared to centers. Do you have an idea of the seasonal variability? Can you test whether those differences could explain a part of your variability?

Line 210: May be add a line saying that the temperature is not the main driver because people will look for this relationship, and that will reinforce your arguments.

The temperatures at the ojos are typically within 1 degree of the lagoon background temperatures and are cooler than the ambient water in summer and warmer in winter. This is because the groundwater has more constant temperatures year-round and the pre-discharge mixing with seawater in the subsurface results in these trends. If the temperature offsets are averaged over a year, then the differences are negligible (typically less than a degree cooler). We adjusted the sentence now in Line 93 to: “... from sites that have similar light conditions differ marginally in temperature (less than 1°C lower at the ojos averaged over all seasons with temperatures cooler than ambient in summer and slightly warmer in winter), have consistently lower salinity (2-4 psu lower than ambient), and are considerably different in Ω_{sw} (supplementary table S1)^{1,37}.”

Hence, temperature is not likely to impact our data since our sampling integrated over a few years of growth. Moreover, as the first reviewer pointed us to recent new publications: At “low pH sites” the temperature is lower than at control sites, since a decrease in pH_{cf} and DIC_{cf} is observed with increasing temperature (Ross, Decarlo, & McCulloch, 2019), temperature in our case can’t explain the decrease of pH_{cf} and DIC_{cf} .” We now also explicitly mention this in the discussion in Line 308: “Environmental factors may also affect pH_{cf} and DIC_{cf} explaining some of the observed differences between the ojo and ambient corals at our study site. Studies have shown that a decrease in pH_{cf} and DIC_{cf} is associated with increasing temperature⁴⁴, yet at our sites the temperatures at the ojos is actually lower, on average, than at control sites.”

The salinity in contrast is indeed consistently lower by 2-4 psu (no seasonal trends). We note that both temperature and salinity were included in calculating the environmental saturation. Clearly since only 41% of the observed lower calcification can be explained by the decrease in Ω_{cf} , it is possible that other factors also contribute, and the lower salinity could be one such factor. Having said that laboratory experiment with the same species at different pH but constant salinity show a decline in calcification (Albright & Langdon, 2011) as we observe, hence the role of salinity cannot be large. However, Ω_{cf} covaries with salinity so it is hard to quantitatively evaluate the salinity effect separately. For an extended model (calcification $\sim \Omega_{cf} + \text{salinity}$) the AIC increased for the simplest model (calcification $\sim \Omega_{cf}$) of 1.74 to 1.80 and slightly failed to be significant for the extended ($p=0.0695$) suggesting Ω_{cf} is the best predictor for the measured calcification response. Regardless we note that previous studies show that corals tended to be more tolerant to salinity than temperature stress (Berkelmans, Jones, & Schaffelke, 2012; Chui & Ang, 2017, and references therein). Although, different corals species may have different tolerance levels to reduced salinity but, in most cases, measured effects are seen only at salinities lower than < 26 psu (Berkelmans et al., 2012; Chui & Ang, 2017; Kerswell & Jones, 2003), which we do not encounter in our study site. We discussed the salinity issue now in Line 312: "Salinity might also influence regulation processes, yet the measured average values (32.2 psu) as well as the salinity range measured (26-36 psu)³⁷ at the springs can be tolerated by corals and the long-term exposure to such conditions may have allowed them to develop mechanisms to better cope and adapt to this variable environment^{65,66}."

7. Line 98: Could the acclimation come from this high Ca content as well?

This is certainly possible, yet we see that our sentence here is a bit misleading. We mentioned that calcium is higher at the seeps when normalized to salinity. In total concentration – which is more relevant for coral calcification – calcium concentration at the ojos is in average similar between control and spring (even at the lower salinity of in average 32.2ppm for all ojo centers as well as the range of average salinity values of 31.1-33.9 ppm). Clearly this similar calcium concentration was not sufficient to mitigate the effect of the lower carbonate ion since the derived internal saturation conditions was still lower at the ojo's. We adjusted this part now in Line 101: "The spring water is characterized by lower pH, higher dissolved inorganic carbon (DIC), higher total alkalinity (TA) but similar calcium concentration compared to the ambient conditions away from the spring influence." We now also included a discussion on calcium regulation following the first reviewers comment that also relates to this questions (see point: 8., 12.).

8. Line 181: This result is central in your paper might be good to explain a bit more at line 439.

We now moved this part more towards the end of the discussion also ending the discussion referring to this result. We now state (Line 288): "The present study also indicates that the acclimation process in different corals encompassed some degree of flexibility in terms of the relative role of pH_{cf} and DIC_{cf} regulation in increasing the Ω_{cf} , with some individuals compensating by adjusting their internal pH_{cf} and others primarily by DIC_{cf} modulation. This may also be true of the role of Ca^{2+}_{cf} upregulation. The relative amount, source and transportation pathways of DIC, H^+ and Ca^{2+} to the site of calcification are still not fully understood⁵⁸ and transport processes may differ between individual corals." And close the discussion with (Line 318): "...Our geochemical model approach assumes steady-state equilibrium conditions; however, the rates of the various transport processes involved in regulating the chemistry of the calcifying fluid will ultimately dictate the calcification response⁶⁷ and these rates may differ

between individual coral genotypes further contributing to the offsets between the model output and observations.”

9. Line 197: “thereby modulating Ω_{cf} and calcification rate”

We adjusted the sentence and added now in Line 184: “...thereby modulating CO_3^{2-} , Ω_{cf} and calcification rate.” Also considering the first reviewers comment.

10. Line 217: I think this paragraph needs to move or towards the conclusion or in the introduction.

We now integrated this paragraph into our conclusion, ending now with (Line 325): “In this study we utilized a dual geochemical proxy approach ($\delta^{11}\text{B}$ & B/Ca) to constrain calcifying fluid carbonate chemistry in *P. astreoides* corals that spent their entire life (decades) under acidified low Ω_{sw} conditions. We found that at the pH_{cf} for corals at the low Ω_{sw} was slightly lower than at the ambient conditions indicating lack achievement of optimal calcification conditions. We also determined that pH_{cf} and DIC_{cf} are independently regulated corroborating the calcification response in *P. astreoides*. The study provides new insights into calcification responses of *P. astreoides* under changing environmental conditions and sheds light on the potential of corals to acclimate^{33,47,52,68,69}. Using the geochemical proxies in combination with the bio-inorganic model brought forward by McCulloch *and colleagues*³³ we could explain 41% of the variability in coral growth rates along a Ω_{sw} gradients. The variability which is not explained indicates that additional physiological and environmental processes contribute to the control of calcification rates in natural environments. This provides promising new avenues towards studying acclimation and adaptation potential of long-lived marine invertebrates such as corals.”

11. Line 262: Is there any data for your corals?

Yes – symbiont densities were higher in corals at low saturation, but symbiont types did not differ. We now integrated this finding into our discussion in Line 296: “Corals at the ojos harbour a higher density of symbionts⁵⁹ that may potentially account for the higher energy demands for pH_{cf} up-regulation resulting in the relatively small difference in the internal conditions (pH_{cf} , DIC_{cf}) we see.”

12. Line 265: So Ω_{cf} explains 44% of the calcification rate variability, all the arguments here are likely to modulate the Ω_{cf} but since it is only 44%, any ideas what are the independent (from the calcifying fluid) triggers for the 56% left?

This is a very good point. We tried to improve our argumentation to make clear (which we did not do in an explicit way) why these parameters can also contribute to the unexplained variation. It is right that they on the one hand will also contribute to the steady-state internal conditions (thus, 41 % explained variability) that we derive from our geochemical data, yet these parameters may also modulate rates of transport processes and thus ultimately the calcification response. We now clearly stated in the revised version that we need to distinguish between these contributions. Another important independent factor that with our approach we could not address is internal calcium regulation. Calcium regulation is certainly another regulating parameter and thus, contribute to the unexplained variation. This argumentation and information is now given in the manuscript. In Line 275 we address the role of calcium regulations: “Recent studies identified internal calcium (Ca^{2+}_{cf}) regulation as an additional player in coral calcification responses and emphasized that regulation of Ca^{2+}_{cf} can contribute to a corals’ resistance to future ocean changes^{42-44,54}. In this sense, the good

agreement of our model with the observed calcification response potentially indicates that internal average steady-state calcium concentrations ($\text{Ca}^{2+}_{\text{cf}}$) are lower at the ojos by some proportion that is related to the pH_{cf} changes, since our model based on pH_{cf} and $\text{CO}_3^{2-}_{\text{cf}}$ can reproduce the observed calcification decline. This suggests a strong link between $\text{Ca}^{2+}_{\text{cf}}$ and pH_{cf} and supports the idea of a plasma-membrane Ca-ATPase^{55, but see 56} responsible for pH_{cf} regulation. However, it is possible that pH_{cf} but also $\text{Ca}^{2+}_{\text{cf}}$ was regulated by additional and/or different ion transport mechanisms (e.g. potentially ion exchangers, Ca^{2+} -channels)^{56,57}. ” Now we moved the other potential contributor to the end of the discussion we close this part (Line 313): “Overall, these environmental and biological parameters may be responsible for the observed internal conditions but likely also affect rates of processes, and thus contribute to the unexplained component in our relation between the calcification and the geochemically derived DIC_{cf} and pH_{cf} .”

13. Line 284: Can you explain why it is controlled by the adjacent epithelium?

During revision of the manuscript, we integrated this part (on DIC concentration and that it lies a certain confined concentration range (Fig. S1) controlled by the adjacent epithelium) into another part (addressing DIC upregulation and the role of the external concentrations levels on the upregulation potential Line 251) changing not only the wording but also deepening the discussion. In general it has been shown that corals aboral epithelia is in tight association with the coral skeleton, and from immunolabelling experiments (Barott, Perez, Linsmayer, & Tresguerres, 2015; Zoccola et al., 2004) that ion-transport associated channels/enzymes/exchangers are located also within this epithelia and allowing the epithelia to regulate transport processes.

14. Line 290: May be add a sentence to conclude more smoothly.

We hope, we did this by restructuring and ending differently in the revised text.

15. Line 320: How did you measure your seawater samples?

Samples were first analyzed on ICP-MS Finnigan Element XR for boron concentration measurements following Krupinski et al. (2017) before they were further analyzed on a Neptune multi-collector inductively coupled mass spectrometer at National Cheng Kung University, Taiwan. The boric acid standard acid standard IAEA-B-1 was used as the reference standard (e.g. 39.77 ‰) to determine the $\delta^{11}\text{B}$ of the samples, reproducibility ($\pm 0.25\%$) using a standard-sample-standard bracketing technique following Wang et al. (2010).

We added this information now at Line 347: “Water samples were also taken for seawater boron concentrations (measured on a ICP-MS Finnigan Element XR following Krupinski and colleagues⁷¹; $\sim 430 \pm 8 \mu\text{M}$, with no difference between ojos and control) and a boron isotopic composition ($\delta^{11}\text{B}_{\text{sw}}$) of 39.15 (1sd = 0.12; n = 3) for the control site and 38.85 (1sd = 0.17; n = 5) for the low pH ojos. Boron isotopic samples were analysed on a Neptune multi-collector inductively coupled mass spectrometer at National Cheng Kung University, Taiwan, using the standard-sample-standard bracketing technique⁷². The boric acid standard IAEA-B-1 was used as the reference standard (e.g. 39.77 ‰) to determine the $\delta^{11}\text{B}$ of the samples, reproducibility ($\pm 0.25\%$).”

16. Line 334: What are those modifications?

We modified the text so that it is clear what were the main modification now in Line 367: “Specifically, we used Multiplier and Faraday cups simultaneously to collect data for B¹⁰ and B¹¹ (both on multiplier) as well as C¹² (Faraday cup). This allows us to derive B/C and δ¹¹B from the same skeletal material. Similar to previous work the cones were cleaned on a regular basis (every 2-4 days). ...”

17. Line 360: Is it due to the natural variability or to the method? Do you have any standards ran in liquid during your session?

The statement in this line refers to the occurrence of two different skeletal enteties in corals and that they are naturally variable. To account for this, we used this approach to make sure that in all corals we cover a similar proportion of both skeletal enteties. We added additional information to make this more clear (Line 389): “By this approach we expect to cover a representative sample set and minimize the natural variability in skeletal enteties and cover similar proportions in the different corals (assuming that COC to fibre ratio in coral grown under various environmental conditions stays constant).”

18. Line 367: Explain quickly B/C, it is confusing with B/Ca even if well explained in the rest of the paragraph. Have you compared your B/Ca with liquid measurements?

B/C elemental ratios have been determined simultaneously with the boron isotope ratios via LA-MC-ICP-MS. Boron isotope data (¹⁰B and ¹¹B) have been collected using a pair of ion counters, while carbon (¹²C) had been determined using a Faraday cup. B/C data are based on the integrated boron intensities (¹⁰B+¹¹B) divided by the ¹²C intensity. The calibration (conversion from intensity ratios to concentration ratios) has been done using a natural *L. pertusa* coral sample covering a B concentration range of about 450-950 μmol/mol, which had been determined before using LA-ICP-MS relative to standard NIST-SRM610 using ⁴³Ca as internal standard. This calibration procedure resulted in: B/C [μmol/mol] = 78800 x B/C [cps/cps]; (cps – counts per second, ion beam intensity). We used stoichiometric ratio of C/Ca = 1 as approximation for natural carbonates and translated B/C ratios in B/Ca [μmol/mol] ratios.

19. Line 439: May be reformulate.

We reformulated these sentence now in Line 482: “Statistical analysis of the geochemical proxies and derived internal calcification conditions was performed by comparing the two treatment groups (control conditions vs. reduced aragonite saturation state at the centers of the ojos) using Welch’s t-test (unequal sample numbers).”

20. Table S2: 1st column, typo 2 decimals

Corrected

References cited:

Albright, R., & Langdon, C. (2011). Ocean acidification impacts multiple early life history processes of the Caribbean coral *Porites astreoides*. *Global Change Biology*, 17(7), 2478–2487.
<https://doi.org/10.1111/j.1365-2486.2011.02404.x>

- Barott, K. L., Perez, S. O., Linsmayer, L. B., & Tresguerres, M. (2015). Differential localization of ion transporters suggests distinct cellular mechanisms for calcification and photosynthesis between two coral species, (44). <https://doi.org/10.1152/ajpregu.00052.2015>
- Berkelmans, R., Jones, A., & Schaffelke, B. (2012). Salinity thresholds of *Acropora* spp. on the Great Barrier Reef. *Coral Reefs*, 31, 1103–1110. <https://doi.org/10.1007/s00338-012-0930-z>
- Chui, A., & Ang, P. J. (2017). High tolerance to temperature and salinity change should enable scleractinian coral *Platygyra acuta* from marginal environments to persist under future climate change. *PLoS ONE*, (12), 1–15. <https://doi.org/10.1371/journal.pone.0179423>
- Kerswell, A. P., & Jones, R. J. (2003). Effects of hypo-osmosis on the coral *Stylophora pistillata*: nature and cause of ' low-salinity bleaching .' *Marine Ecology Progress Series*, 253(1964), 145–154.
- Lee, K., Kim, T., Byrne, R. H., Millero, F. J., Feely, R. A., & Liu, Y. (2010). The universal ratio of boron to chlorinity for the North Pacific and North Atlantic oceans. *Geochimica et Cosmochimica Acta*, 74(6), 1801–1811. <https://doi.org/10.1016/j.gca.2009.12.027>
- Ross, C. L., Decarlo, T. M., & Mcculloch, M. T. (2019). Environmental and physiochemical controls on coral calcification along a latitudinal temperature gradient in Western Australia. *Global Change Biology*, 25(August), 431–447. <https://doi.org/10.1111/gcb.14488>
- Wang, B.-S., You, C.-F., Huang, K.-F., Wu, S.-F., Aggarwal, S. K., Chung, C.-H., & Lin, P.-Y. (2010). Direct separation of boron from Na- and Ca-rich matrices by sublimation for stable isotope measurement by MC-ICP-MS. *Talanta*, 82(4), 1378–1384. <https://doi.org/10.1016/j.talanta.2010.07.010>
- Zoccola, D., Tambutté, E., Kulhanek, E., Puverel, S., Scimeca, J.-C., Allemand, D., & Tambutté, S. (2004). Molecular cloning and localization of a PMCA P-type calcium ATPase from the coral *Stylophora pistillata*. *Biochimica et Biophysica Acta*, 1663(1–2), 117–26. <https://doi.org/10.1016/j.bbamem.2004.02.010>

REVIEWERS' COMMENTS:

Reviewer #1 (Remarks to the Author):

The authors did a nice job of revising their manuscript to address all the points raised in the initial reviews. I think the revised manuscript has improved substantially and it will make a strong contribution to the literature. I did notice quite a few minor grammatical errors though, so the manuscript should receive a final careful check for grammar.

- Thomas DeCarlo

REVIEWERS' COMMENTS:

Reviewer #1 (Remarks to the Author):

The authors did a nice job of revising their manuscript to address all the points raised in the initial reviews. I think the revised manuscript has improved substantially and it will make a strong contribution to the literature. I did notice quite a few minor grammatical errors though, so the manuscript should receive a final careful check for grammar.

- Thomas DeCarlo

Authors response:

We thank the reviewer to help improve our manuscript and revised our manuscript by doing a thorough check of grammar.